# A General Aggregation Federated Learning Intervention Algorithm based on $do$-Calculus

## Abstract

This article explores federated long-tail learning (Fed-LT) tasks, where clients possess private, heterogeneous data, collectively following a global long-tail distribution. We propose two methods: (a) Client Re-weighted Prior Analyzer (CRePA), which balances the global model's performance on tail and non-tail categories and enhances performance on tail categories while maintaining it on non-tail categories. (b) Federated Long-Tail Causal Intervention Model (FedLT-CI) computes clients' causal effects on the global model's performance in the tail and enhances the interpretability of Fed-LT. Extensive experiments on the CIFAR-10-LT and CIFAR-100-LT datasets demonstrate the following: (1) CRePA outperforms other baselines, achieving state-of-the-art (SOTA) performance. In scenarios with high heterogeneity and severe long-tail distributions, CRePA improves tail performance by 6.3% and 5% compared to CReFF and FedGrab, respectively. (2) FedLT-CI, by intervening during the aggregation process in federated learning (FL), effectively enhances the tail performance of baselines while maintaining stable non-tail performance. For instance, applying the intervention strategy to the FedAvg, FedGrab, and CRePA models improves tail performance by 4.5%, 2.1%, and 1.9%, respectively.

## 1 Introduction

The issue of tackling the long-tail problem of client data and server's aggregated data in FL has emerged as a crucial concern in algorithm design. Unlike traditional machine learning, FL brings the learning task directly to the end-user devices for local training, requiring only intermediate parameters (such as gradients) to be sent to the server for model aggregation and updates. This approach contributes to obtaining a globally applicable model while preserving the privacy of client-side data, facilitating the development of trustworthy intelligent systems (Xiao et al., 2023). Despite the great potential of FL, its real-world applications still face numerous challenges. One significant challenge is the heterogeneity of data among clients, i.e., Non-IID (Li et al., 2022b; Xu et al., 2022; Li et al., 2023; Tang et al., 2024). Additionally, while the assumption is that the distribution of data sets for clients may be locally balanced, aggregating the data from various clients might result in a severe long-tail distribution issue, named the Fed-LT (Xiao et al., 2023; Zeng et al., 2023; Shang et al., 2022). For example, there may be significant differences in data between different hospitals, and different types of diseases exhibit a severe long-tail distribution (Caesar et al., 2020). Although the tail categories are less common in a long-tail distribution, they are crucial for identifying rare diseases(Zhang et al., 2023), dangerous behaviors in autonomous driving (Wang et al., 2022), and more.

In the FL scenario, if the training data of different clients exhibit long-tailed and heterogeneous characteristics, the problem becomes complex and challenging because each client may contain different categories and quantities of tail data. This situation significantly impacts the performance of the global model, especially in tail categories. Currently, algorithms used to address data heterogeneity in FL ignore the potential long-tailed issues (McMahan et al., 2017; Karimireddy et al., 2020; Luo et al., 2021). However, if one directly adopts some long-tail learning algorithms to address the long-tail problem in FL, local or global class distribution information is needed as prior knowledge for optimization, which may expose potential privacy concerns (Shang et al., 2022). For example, some methods typically rely on statistical information from client data, such as the sample count or feature distribution of different classes, as well as global class distribution information.

Some researchers have proposed solutions to the heterogeneity and long-tail problems in FL (Shang et al., 2022; Xiao et al., 2023). These methods have shown significant improvements in tail performance, but they all come with increased communication costs. Importantly, none of these methods consider the impact of different clients on the aggregation model's performance on tail data from a causal perspective, and they lack interpretability.

Given the challenges above, this paper addresses the long-tail and heterogeneity issues in FL by proposing two novel models to enhance tail performance: Client Re-weighted Prior Analyzer and the Federated Long-Tail Causal Intervention Model. The CRePA learns the prior distribution of weights for each client through tail and non-tail gradient information (as shown in Fig. 2). Thus, it allows flexible balancing between tail and non-tail data in the heterogeneous datasets of different clients, thereby improving the global model's performance on tail categories.

In order to address the long-tail challenges more comprehensively in FL, we propose a novel and general FedLT-CI model inspired by the do-operator proposed by Pearl (1995) (as shown in Fig. 4). This model improves the traditional FL aggregation process by introducing an intervention mechanism, significantly enhancing tail performance (as shown in Fig. 1). The FedLT-CI can discern the causal effects of each client on the server-aggregated model regarding tail data. This model enhances the overall model's performance on tail data and possesses interpretability (for instance, it can be understood as likening the performance of the aggregated model to a disease, treating client participation as a form of treatment, and evaluating its causal effect to measure the treatment outcome).

Our main contributions are summarized as follows.

1. We propose CRePA, a client-weighted sampling algorithm that uses Monte Carlo sampling to dynamically allocate weights, addressing heterogeneity and long-tail challenges in FL. Without relying on prior client knowledge, CRePA employs an adaptive loss function to seamlessly integrate tail and non-tail gradients, enabling online learning of weight parameters for flexible model adaptation. CRePA outperforms SOTA models, especially in enhancing tail data performance.

2. We propose FedLT-CI, a model that enhances tail performance in FL by leveraging causal interventions. FedLT-CI assesses clients' causal effects on the global model's tail performance and adjusts intervention frequency to mitigate the impact of weaker contributors. This adaptable mechanism integrates seamlessly with most FL algorithms. Experiments demonstrate that FedLT-CI significantly improves tail performance while maintaining non-tail performance across various baselines.

3. Without compromising algorithm performance, our algorithm utilizes fewer clients in the information aggregation process due to the introduction of causal inference. This ingenious design improves the model's performance on the tail data and reduces communication overhead. In this way, a core issue of FL algorithms, namely security, has been indirectly improved.

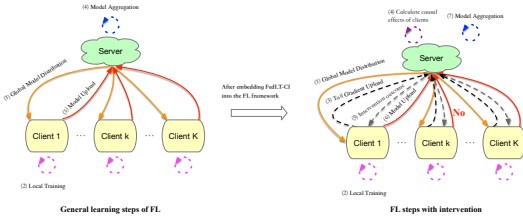

Figure 1: Comparison Before and After Embedding Interventions in Federated Learning. In step (7) of integrating FedLT-CI into the FL framework, we can use baseline models, including CRePA.

## 2 RELATED WORK

### 2.1 FEDERATED LEARNING WITH DATA HETEROGENEITY

Researchers have proposed various methods to investigate the challenge of heterogeneous or imbalanced data distribution in FL (Xiao et al., 2024). To address the challenge of heterogeneous or imbalanced data distribution in FL, researchers have proposed various methods, such as adding a regularizer locally to modify the local loss function (Li et al., 2020; Durmus et al., 2021). Karimireddy et al. (2020) proposed a control variable-based method to reduce client distribution drift

caused by differences in data distribution. CCVR samples virtual features from an approximated Gaussian Mixture Model for classifier calibration to avoid uploading raw features to the server (Luo et al., 2021). However, the above methods neglect the global long-tail distribution, leading to poor performance on tail classes.

## 2.2 LONG-TAILED LEARNING

Due to the widespread presence of long-tail data in the real world, long-tail learning in machine learning has garnered considerable attention from various researchers (Zhang et al., 2023; Wang et al., 2024; Zhang et al., 2024; Narasimhan et al., 2024). Methods include techniques that re-weight based on the frequencies of different classes (Cao et al., 2019) or utilize information augmentation techniques to enhance the performance of tail categories (Li et al., 2021). Among these, re-weighting methods aim to improve the model's performance in tail categories by balancing losses or gradients. In addition, some methods decouple the training phase into representation learning and classifier re-training (Kang et al., 2020; Shang et al., 2022). These methods aim to generate more general representations and enhance the performance of tail categories on a re-balanced classifier. However, most of the methods mentioned rely on the global class distribution. During FL's training process, collecting information on the class distribution from each client is impractical to obtain the global class distribution. This renders the majority of long-tail learning methods unsuitable for the FL scenario.

## 2.3 CAUSAL EFFECT

Some researchers directly learn causal effects from local data (Alaa & Schaar, 2018; Künzel et al., 2019; Yao et al., 2018). Others use the Structural Causal Model (SCM) proposed by Pearl (1995) to estimate the causal effects involving latent confounding variables (Madras et al., 2019; Kawakami et al., 2023; Wang et al., 2023; Shirahmad Gale Bagi et al., 2023). Shirahmad Gale Bagi et al. (2023) designed a causal model to address the motion prediction task. In this paper, we employ the SCM method to calculate the causal effects of clients on the global model's performance in the tail. Unlike previous methods for causal effect estimation, which typically require access to all data, in FL, we cannot directly aggregate all client data, posing unique challenges (Li et al., 2024). To the best of our knowledge, no solutions have been designed from a causal effects perspective to address the challenges of Fed-LT. Therefore, the method proposed in this paper represents a novel attempt to tackle the issues associated with Fed-LT, and the experimental results validate the effectiveness of this approach.

# 3 PROPOSED METHODS

## 3.1 PRELIMINARIES

**Federated Learning:** Assuming we have $K$ clients, each denoted as $k \in [K]$, with data categorized into tail and non-tail classes, represented as $D^k = \{D^k_{nott}, D^k_t\} = \{d^k_{nott_1}, d^k_{nott_2}, \cdots, d^k_{nott_i}, d^k_{t_{i+1}} \cdots, d^k_{t_n}\}$. Here, $D^k_{nott}$ represents the non-tail data for client $k$. $D^k_t$ represents the tail data for client $k$, and $|D^k|$ indicates the size of the data. These data sources $D^k$ belong to different clients, and their distributions may be entirely different. FL aims to achieve convergence of the global model through multiple rounds of communication with clients. Clients locally train models, and the results are aggregated on the server.

**Definition of Tail gradient and Non-tail gradient:** In tasks such as visual recognition (Tan et al., 2020) and object detection (Tan et al., 2021), gradient information is crucial in addressing long-tail issues. Unlike these tasks that use gradient information, we further classify gradients into two types: tail and non-tail. Considering a classification model based on a neural network, with the cross-entropy loss function denoted as $\mathcal{L}$, we represent the parameters of the last layer classifier of the model as $\mathbf{w} = [w_1, w_2, \cdots, w_C]$, where $C$ represents the number of classes, $w_i = [w_{i1}, w_{i2}, \cdots, w_{id}]$ denotes the parameters for class $i$ and $d$ is the size of the neurons in the preceding layer. After deriving the loss function $\mathcal{L}$ with respect to $\mathbf{w}$, the gradient information on the parameters for class $i$ is represented as $\nabla_{w_i}(\mathcal{L}) = [\nabla_{w_{i1}}(\mathcal{L}), \nabla_{w_{i2}}(\mathcal{L}), \cdots, \nabla_{w_{id}}(\mathcal{L})]$. If class $i$ belongs to the tail, the gradients for this class are labeled as tail gradients. This differentiation be-

tween tail and non-tail gradients helps adopt distinct optimization strategies for addressing long-tail issues, enabling the model to learn and adapt to the features of tail classes more effectively.

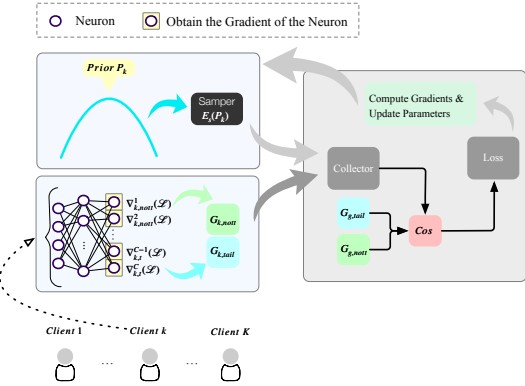

Figure 2: CRePA Online Learning Process.

## 3.2 CLIENT RE-WEIGHTED PRIOR ANALYZER (CRePA)

**Prior** $P_k$**:** $P_k$ represents the prior distribution of weights for client $k$, with parameters $w_{pk}$ initialized and updated by the server. During model aggregation, the expected value $\mathcal{E}_s(P_k)$ is approximated using Monte Carlo sampling, and this is used as the weight for client $k$. As a self-adjusting balancer, we employ a Gaussian distribution to construct the prior distribution, addressing the challenges of long-tail and heterogeneous data. The specific learning process is illustrated in Fig. 2.

**Collector:** In the process of learning distribution parameters, the primary role of the collector is to gather gradient information (i.e., $G_{k,nott}$, $G_{k,tail}$) from client $k \in [K]$, and approximate the expectation $\mathbb{E}_s(P_k)$ through Monte Carlo sampling.

$$G_{k,nott} = \eta_k \nabla_{W_{k,nott}}(\mathcal{L}) = \eta_k (\nabla_{w_{k,1}}(\mathcal{L}) \cdots \nabla_{w_{k,j}}(\mathcal{L}))^T = \eta_k \nabla \begin{pmatrix} w_{1,1} & w_{1,2} & \cdots & w_{1,d} \\ w_{2,1} & w_{2,2} & \cdots & w_{2,d} \\ \vdots & \vdots & \vdots & \vdots \\ w_{j,1} & w_{j,2} & \cdots & w_{j,d} \end{pmatrix}.$$

(1)

$G_{k,nott}$ implies the non-tail gradients information of client $k$, $\mathcal{L}$ represents the loss of the local training classification model on the client, $\eta_k$ represents the learning rate of client $k$, $\nabla_{W_{k,nott}}(\mathcal{L})$ indicates the gradients of client $k$ on the classification layer parameters for non-tail classes, and $j$ represents the number of non-tail categories.

$$G_{k,tail} = \eta_k \nabla_{W_{k,tail}}(\mathcal{L}) = \eta_k (\nabla_{w_{k,1}}(\mathcal{L}) \cdots \nabla_{w_{k,i}}(\mathcal{L}))^T = \eta_k \nabla \begin{pmatrix} w_{1,1} & w_{1,2} & \cdots & w_{1,d} \\ w_{2,1} & w_{2,2} & \cdots & w_{2,d} \\ \vdots & \vdots & \vdots & \vdots \\ w_{i,1} & w_{i,2} & \cdots & w_{i,d} \end{pmatrix}.$$

(2)

$$\mathbb{E}_s(P_k) = \frac{1}{M} \sum_{x_i \sim P_k}^{M} x_i.$$

(3)

$G_{k,tail}$ denotes the tail gradients of client $k$, $\nabla_{W_{k,tail}}(\mathcal{L})$ denotes the gradients of client $k$ on the classification layer parameters for tail classes, and $i$ denotes the number of tail categories. $M$ is the sampling size, and $x_i$ denotes the sampled value from $P_k$. In Fig. 2, the function of $Cos$ is to calculate the cosine similarity between the client gradients (i.e., $G_{k,nott}$, $G_{k,tail}$) and

the global gradients (i.e., $G_{g,nott}$, $G_{g,tail}$). Define $S_{k,nott} := \eta_k \sum_{l \notin [tail]} \frac{\nabla_{w_{k,l}} \cdot \nabla_{w_{g,l}}}{\|\nabla_{w_{k,l}}\| \|\nabla_{w_{g,l}}\|}$ and $S_{k,tail} := \eta_k \sum_{l \in [tail]} \frac{\nabla_{w_{k,l}} \cdot \nabla_{w_{g,l}}}{\|\nabla_{w_{k,l}}\| \|\nabla_{w_{g,l}}\|}$ as the sums of the similarity scores between the client's non-tail gradients and tail gradients, respectively.

**CRePA Loss Function Design:** First, we define the set $\mathcal{B}$ as the weighted sum of client $\{S_{k,not}, S_{k,tail}\}$:

$$\mathcal{B} := \{\beta S_{k,tail} + (1 - \beta) S_{k,nott}\}_{i=1}^K \tag{4}$$

Where $\beta$ is the weight focusing on the tail gradient, a larger weight indicates a greater emphasis on the tail information of each client when learning the prior distribution comprehensively. Sort the set $\mathcal{B}$ in descending order, we get: $\mathcal{B}', \mathcal{B}_{(1,x_1)} \geq \cdots \geq \mathcal{B}_{(i,x_i)} \geq \mathcal{B}_{(i+1,x_{i+1})} \geq \cdots \geq \mathcal{B}_{(K,x_K)}$. Where $x_i$ represents the index of the client with identifier $i$. Let $\Delta_i = \mathbb{E}_s(P_{x_i})\mathcal{B}_{(i,x_i)} - \mathbb{E}_s(P_{x_{i+1}})\mathcal{B}_{(i+1,x_{i+1})}$, The loss $\mathcal{L}_{prior}$ is designed as follows:

$$\mathcal{L}_{prior} = \sum_{i=1}^{K-1} \Delta_i. \tag{5}$$

Therefore, we update the prior distribution parameters $w_{pk}$ for each client using the designed $\mathcal{L}_{prior}$ function. **We present the full algorithm in Appendix B.1.**

## 4 FEDERATED LONG-TAIL CAUSAL INTERVENTION MODEL (FEDLT-CI)

### 4.1 PROBLEM DESCRIPTION

**Problem setting & notations:** In the case where the client data sources $D^k$ have completely different distributions, our goal is to develop a global causal intervention model that satisfies the following two conditions: (i) The causal intervention model is trained in a process where information from each source is not shared with external parties, and the server is not aware of any client data information in advance, (ii) During the iteration process, the causal intervention model can estimate the causal effects of each client on the global model's performance on the tail. The model is illustrated in Fig. 3, and its role is to intervene in the aggregation process through causal effects, allowing the aggregated model to perform better on tail data.

**Causal effects:** Given the causal model trained under the above settings, we consider the performance of the global model on tail data as the outcome ($Y$), When intervening on client $k \in [K]$, this client is treated as the treatment, while the remaining clients serve as confounding factors. In a loose sense, our interest can be viewed as estimating the individual treatment effect (ITE). In this scenario, treating each client as an individual, every client can be considered a target for intervention. Our primary goal is to estimate ITE, i.e., the causal effect on the global model's tail performance when client $k$ participates or does not participate. This can be expressed using the following formula:

$$\tau := \mathbb{E}[Y|do(c_k = 1), c_1, \cdots, c_{k-1}, c_{k+1}, \cdots, c_K] \\ - \mathbb{E}[Y|do(c_k = 0), c_1, \cdots, c_{k-1}, c_{k+1}, \cdots, c_K] \tag{6}$$

We use the symbol $\tau$ to represent the expectation of the outcome $Y$ when intervening on client $k$, which is also the core of our task. Where $do(c_k = 1)$ denotes the intervention of client $k$ participating in the aggregation of the server-side model, and $do(c_k = 0)$ indicates that client $k$ does not participate. This utilizes Pearl's $do$-calculus operation (Pearl, 1995) to calculate causal effects.

In the causal graph, $Y'$ is defined as follows. Assuming the current iteration is the $t$-th iteration, $Y'$ represents the cosine similarity vector between the tail gradients of each client in the $t$-th iteration and the aggregated tail gradient at the $t - 1$ time step.

$$Y' = [Y'_1, Y'_2, \cdots, Y'_K]^T, where\ Y'_{k,k\in[K]} = \frac{1}{N_{C_{tail}}} \sum_{l \in [tail]} \frac{\nabla_{w_{k,l}} \cdot \nabla_{w_{g,l}}}{\|\nabla_{w_{k,l}}\| \|\nabla_{w_{g,l}}\|}. \tag{7}$$

### 4.2 LEARNING LATENT VARIABLES THROUGH VARIATIONAL INFERENCE

According to Fig. 3, there are $K$ potential confounding factors that can influence the performance of the global model on tail data. Therefore, based on the causal graph, we can use $do$-calculus to

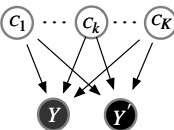

Figure 3: Graphical structure of the proposed FedLT-CI. Where $C_k$ represents the client confounding factor, $Y$ denotes the outcome (performance of the aggregated model on tail data), and $Y^{'}$ represents the tail gradient information from clients during iterations, as seen in Equation 7. Solid circles denote observed variables, while hollow shapes represent unobserved variables.

compute the causal effect of client $k$ on the model:

$$p(Y|do(c_i=w)) = \int p(Y|c_1,\cdots,c_i=w\cdots,c_K,Y^{'}) * p(c_1|Y^{'})\cdots p(c_K|Y^{'})dY^{'}$$
$$= \mathbb{E}_{p(c_1|Y^{'}),\cdots,p(c_K|Y^{'})}\left[p(Y|c_1,\cdots,c_i=w\cdots,c_K)\right]. \tag{8}$$

Where $w \in \{0,1\}$. Eq. (8) indicates that if we can find each client, the causal effect is identifiable. The second step is to use the backdoor adjustment formula. Although we do not know the true posterior distributions of latent variables $(c_1, c_2, \cdots, c_K)$, we can use mean variational inference to approximate them (Goodfellow et al., 2016).

Standard logarithmic likelihood function is:

$$\max_{p} \mathbb{E}_{p^*(Y,Y^{'})}\left[\log p(Y,Y^{'})\right]. \tag{9}$$

Here, $p^*(Y,Y^{'})$ is the true joint distribution of $Y$ and $Y^{'}$, but computing $\log p(Y,Y^{'})$ is challenging, as we only have access to a small number of samples in each communication round. However, mean-field variational inference can be employed to approximate the true posterior distribution of latent variables, providing us with an approach to address this issue.

We use the *Evidence Lower Bound* (ELBO) as the objective function to train the model, which is given by:

$$\max_{p,q} \mathbb{E}_{p^*(Y,Y^{'})}\mathbb{E}_{q(c_1,c_2,\cdots,c_K|Y,Y^{'})}\left[\log \frac{p(c_1,c_2,\cdots,c_K,Y,Y^{'})}{q(c_1,c_2,\cdots,c_K|Y,Y^{'})}\right] \tag{10}$$

In theory, the ELBO function will guide $q(c_1,c_2,\cdots,c_K|Y,Y^{'})$ towards the target $p(c_1,c_2,\cdots,c_K|Y,Y^{'})$. To train the causal model further, the loss function in Equation 10 is further transformed as:

$$\max_{p} \max_{q} \mathbb{E}_{p^*(YY^{'})}\left[\log q(Y)\right] + \mathbb{E}_{p^*(YY^{'})}\mathbb{E}_{q(c_1,c_2,\cdots,c_K|Y^{'})}$$
$$\left[\frac{q(Y|c_1,c_2,d\cdots,c_K,Y^{'})}{q(Y)}\log \frac{p(Y^{'}|c_1,c_2,\cdots,c_K)p(c_1)p(c_2)\cdots p(c_K)}{q(c_1|Y^{'})q(c_2|Y^{'})\cdots q(c_K|Y^{'})}\right] \tag{11}$$

The derivation from Equation 10 to Equation 11 can be referred to in Appendix A. Additionally, on top of the mentioned loss, we introduce a penalty term: $\frac{1}{K}\sum_{i=1}^{K}\left(p(Y|do(c_i=1)) - p(Y|do(c_i=0)) - Y_k^{'}\right)^2$. The purpose of this penalty term is to consider the corresponding impact of the gradient information from client $k$ in the current iteration when calculating the causal effect of client $k$. The overall overview of our model is shown in Fig. 4. We calculate $Y^{'}$ by processing the tail gradient information from each client, followed by interaction and modeling to approximate the posterior distribution of each client. Simultaneously, we sample from the approximate posterior distribution and interact to reconstruct the information $Y^{'}$. We use a Gaussian Mixture Models (GMMs) to model the prior distribution $P(c_k)$ of each client. The GMMS is considered a universal approximator, possessing powerful modeling capabilities to flexibly capture and generate various variant features. Therefore, it is frequently employed in models such as VAE (Jiang et al., 2017; Shirahmad Gale Bagi et al., 2023). We chose to use a Gaussian Mixture Prior based on the research by Kivva et al. (2022), who

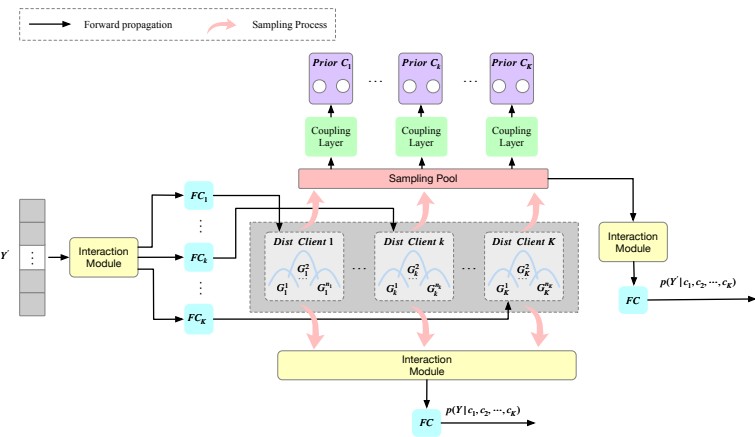

Figure 4: The overall overview of the FedLT-CI

demonstrated the identifiability of variational models with a GMM prior. In the model, we also borrowed the idea of coupling layers (Dinh et al., 2016; Shirahmad Gale Bagi et al., 2023) to learn rich priors $P(c_k)$.

**Intervention process:** As the communication iteration progresses, we intervene at a certain frequency $\mathcal{F}$ during the aggregation process. The process of each intervention is as follows: Calculate the causal effect of each client on the global model based on the formula (Eq. 6). Then, select a certain number $\mathcal{N}$ of clients with lower causal effects, making them excluded from the aggregation.

**The detailed process of applying FedLT-CI to the baseline is introduced in Appendix B.2.**

### 4.3 COMMUNICATION COST ANALYSIS:

In the intervention model, with intervention frequency $\mathcal{F}$, communication rounds $T$, client count $K$, and intervention client count $N$, the communication cost of applying FedLT-CI can be reduced to $1 - \frac{N}{2TK}\left\lfloor\frac{T}{\mathcal{F}+1}\right\rfloor$ times the original cost (i.e., $\frac{N}{2TK}\left\lfloor\frac{T}{\mathcal{F}+1}\right\rfloor$ represents the percentage of communication cost reduction after intervention). For detailed proof, please refer to the Appendix A.2.

## 5 EXPERIMENTS

### 5.1 EXPERIMENTAL SETUP

**Baselines:** We adopt two SOTA baselines for comparative experiments. Firstly, targeting data heterogeneity, we chose the FedAvg (McMahan et al., 2017) and FedProx (Li et al., 2020) methods. Secondly, focusing on long-tail learning, we selected the CReFF (Shang et al., 2022), GCL-loss (Li et al., 2022a), DisA (Gao et al., 2024), and Fed-GraB (Xiao et al., 2023) methods.

**Datasets:** We conduct long-tail classification experiments on three benchmark datasets, i.e., CIFAR-10/100-LT (Krizhevsky & Hinton, 2009) and PTB-XL (Wagner et al., 2020). To simulate a long-tail distribution, we reshape the originally balanced the datasets into a long-tail distribution with IF = 100, 50, and 10. IF represents the degree of the long-tail distribution, defined as the ratio of the number of training samples in the largest class to that in the smallest class. We used the Dirichlet distribution (the parameter $\alpha$ quantifies the Non-IID degree) to generate heterogeneous data partitions among clients (Lin et al., 2020). On CIFAR-10/100-LT, we set the values of $\alpha$ to 0.5 and 1. In the publicly available PTB-XL electrocardiography dataset, we set the value of $\alpha$ to 0.1. In the tests, we further divided the head data into Head and Medium, while representing the tail data as Few, to provide a more detailed demonstration of the model's effectiveness.

**Federated Learning Setting:** During training on the CIFAR-10-LT dataset, we use the ResNet18 model and set the number of clients $K = 40$, $\mathcal{F} = 2$ and $\mathcal{N} = 10$. When training on the CIFAR-100-LT dataset, we use the ResNet34 model and set the number of clients $K = 20/40$, $\mathcal{F} = 2$ and $\mathcal{N} =$

5. For fair comparison, we implemented all FL methods using the same model. All experiments were conducted on the PyTorch framework and executed on an *NVIDIA GeForce RTX 3090 GPU*. We utilized standard cross-entropy as the client loss function, running for 300 communication rounds. SGD was chosen as the optimizer for all optimization processes with a learning rate of 0.01.

## 5.2 COMPARISON WITH STATE-OF-THE-ART METHODS

**Evaluation on CIFAR-10-LT.**

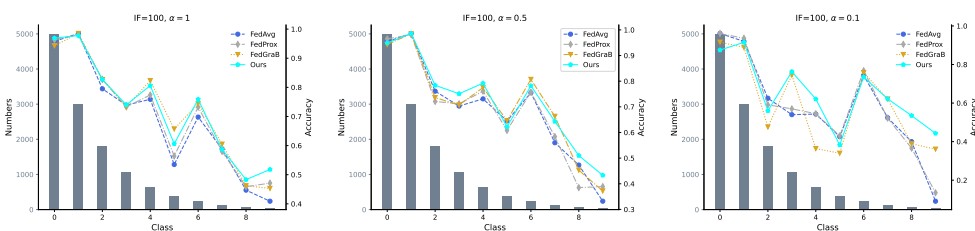

(a) CIFAR-10-LT with the settings of $IF_G$=100, $\alpha$=1, $K$=40).

(b) CIFAR-10-LT with the settings of $IF_G$=100, $\alpha$=0.5, $K$=40).

(c) CIFAR-10-LT with the settings of $IF_G$=100, $\alpha$=0.1, $K$=40).

Figure 5: Without causal intervention, the testing accuracy of CRePA was compared with baselines across different categories, with the last four categories being tail classes.

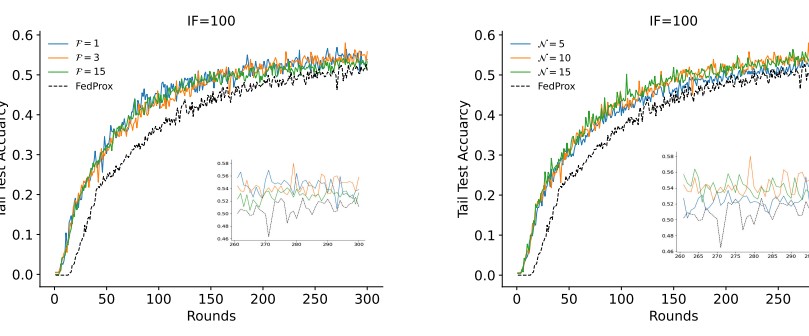

(a) Performance of tail under different $\mathcal{F}$.

(b) Performance of tail under different values of $\mathcal{N}$.

Figure 6: Ablation experiments under FedProx algorithm. The black dashed line represents the performance without our causal intervention algorithm ($\alpha = 0.5, IF_G = 100$). To highlight the differences between the curves, we zoom in on the results of the last few rounds in the top left corner of the figure.

**CRePA's Performance:** Table 1 depicts the experimental results for all SOTA baselines under three different Non-IID settings and three different IF settings. Two main results can be derived from the table: (1) Without considering causal intervention, our CRePA algorithm maintains good performance on head categories while outperforming other algorithms on tail data (For example, in the case of $\alpha$=0.5 and $IF_G$=100, the tail performance is improved by 6%, 6.2%, 4.8% compared to FedAvg, FedProx, and CReFF, respectively. In the case of $\alpha$=0.1 and $IF_G$=100, the tail performance is improved by 14.4%, 13.6%, 6.3%, 5% compared to FedAvg, FedProx, CReFF, and Fed-GraB, respectively.); (2) In most cases, the overall performance of the CRePA algorithm is superior to other baselines. For instance, under the settings of $\alpha$=0.5 and $IF_G$=100, the highest overall test accuracy reaches 72.5%.

For a more in-depth analysis of CRePA's performance on tail data, we compared the test accuracy achieved by different baselines in each category and visualized the results in Fig. 5. It is evident that our model performs better, particularly in tail data and categories 8 and 9, achieving optimal results.

Table 1: Top 1 test accuracies of our methods and SOTA methods on CIFAR-10-LT with diverse imbalanced and heterogeneous data settings. + indicates the incorporation of our intervention algorithm in the baseline aggregation task. Bold: best results. Underlined: the best results without intervention.

| SETTING | METHOD | $IF_G = 10$ | | | | $IF_G = 50$ | | | | $IF_G = 100$ | | | |
| --- | --- | --- | --- | --- | --- | --- | --- | --- | --- | --- | --- | --- | --- |
| | | MANY | MED | FEW | ALL | MANY | MED | FEW | ALL | MANY | MED | FEW | ALL |
| $\alpha=1$ | FedAvg | 0.968 | 0.833 | 0.870 | 0.875 | 0.974 | 0.764 | 0.691 | 0.777 | 0.970 | 0.714 | 0.540 | 0.696 |
| | FedAvg+ | 0.959 | 0.831 | $0.882_{+.012}$ | 0.877 | 0.970 | 0.764 | $0.709_{+.018}$ | 0.783 | 0.972 | 0.723 | $0.580_{+.040}$ | 0.716 |
| | FedProx | 0.962 | 0.827 | 0.863 | 0.869 | 0.969 | 0.771 | 0.682 | 0.775 | 0.972 | 0.724 | 0.560 | 0.708 |
| | FedProx+ | 0.960 | 0.821 | $0.876_{+.013}$ | 0.871 | 0.970 | 0.773 | $0.698_{+.017}$ | 0.782 | 0.973 | 0.731 | $0.583_{+.023}$ | 0.720 |
| | CReFF | 0.962 | 0.835 | 0.873 | 0.876 | 0.951 | 0.734 | 0.672 | 0.753 | 0.962 | 0.721 | 0.554 | 0.702 |
| | CReFF+ | 0.966 | 0.836 | $0.878_{+.005}$ | 0.879 | 0.955 | 0.743 | $0.679_{+.007}$ | 0.760 | 0.963 | 0.731 | $0.577_{+.023}$ | 0.716 |
| | GCL-FL | 0.940 | 0.757 | 0.873 | 0.840 | 0.962 | 0.783 | 0.721 | 0.794 | 0.957 | 0.731 | 0.584 | 0.717 |
| | GCL-FL+ | 0.932 | 0.758 | $0.885_{+.012}$ | 0.844 | 0.950 | 0.793 | $\mathbf{0.742}_{+.021}$ | 0.804 | 0.959 | 0.740 | $\mathbf{0.608}_{+.024}$ | 0.731 |
| | Fed-GraB | 0.948 | 0.834 | 0.882 | 0.876 | 0.962 | 0.793 | 0.708 | 0.793 | 0.964 | 0.760 | 0.576 | 0.727 |
| | Fed-GraB+ | 0.949 | 0.829 | $0.891_{+.009}$ | 0.876 | 0.965 | 0.786 | $0.730_{+.022}$ | 0.799 | 0.964 | 0.758 | $0.597_{+.021}$ | 0.735 |
| | CRePA | 0.960 | 0.839 | 0.887 | 0.883 | 0.968 | 0.783 | 0.728 | 0.798 | 0.970 | 0.737 | 0.593 | 0.726 |
| | CRePA+ | 0.963 | 0.836 | $\mathbf{0.893}_{+.006}$ | 0.884 | 0.970 | 0.787 | $0.721_{+.007}$ | 0.797 | 0.971 | 0.750 | $0.591_{-.002}$ | 0.731 |
| $\alpha=0.5$ | FedAvg | 0.965 | 0.806 | 0.868 | 0.863 | 0.975 | 0.744 | 0.683 | 0.766 | 0.972 | 0.709 | 0.529 | 0.690 |
| | FedAvg+ | 0.958 | 0.818 | $0.881_{+.013}$ | 0.871 | 0.974 | 0.752 | $0.704_{+.021}$ | 0.778 | 0.970 | 0.713 | $0.561_{+.032}$ | 0.704 |
| | FedProx | 0.960 | 0.825 | 0.868 | 0.869 | 0.969 | 0.748 | 0.684 | 0.767 | 0.973 | 0.700 | 0.527 | 0.686 |
| | FedProx+ | 0.964 | 0.833 | $0.877_{+.009}$ | 0.878 | 0.969 | 0.752 | $0.701_{+.017}$ | 0.775 | 0.967 | 0.723 | $0.575_{+.048}$ | 0.713 |
| | CReFF | 0.947 | 0.786 | 0.872 | 0.853 | 0.952 | 0.734 | 0.668 | 0.751 | 0.945 | 0.712 | 0.541 | 0.690 |
| | CReFF+ | 0.951 | 0.790 | $0.885_{+.013}$ | 0.860 | 0.953 | 0.736 | $0.686_{+.018}$ | 0.759 | 0.942 | 0.720 | $0.559_{+.018}$ | 0.700 |
| | GCL-FL | 0.932 | 0.813 | 0.871 | 0.860 | 0.959 | 0.759 | 0.712 | 0.780 | 0.959 | 0.709 | 0.571 | 0.704 |
| | GCL-FL+ | 0.927 | 0.812 | $0.890_{+.019}$ | 0.866 | 0.961 | 0.763 | $0.727_{+.015}$ | 0.788 | 0.955 | 0.706 | $\mathbf{0.605}_{+.034}$ | 0.716 |
| | Fed-GraB | 0.932 | 0.813 | 0.887 | 0.866 | 0.946 | 0.764 | 0.711 | 0.779 | 0.964 | 0.716 | 0.581 | 0.712 |
| | Fed-GraB+ | 0.937 | 0.807 | $\mathbf{0.896}_{+.009}$ | 0.869 | 0.949 | 0.783 | $\mathbf{0.731}_{+.020}$ | 0.796 | 0.960 | 0.735 | $0.596_{+.015}$ | 0.724 |
| | CRePA | 0.957 | 0.821 | 0.889 | 0.875 | 0.964 | 0.762 | 0.721 | 0.786 | 0.969 | 0.739 | 0.589 | 0.725 |
| | CRePA+ | 0.955 | 0.823 | $0.891_{+.002}$ | 0.877 | 0.965 | 0.776 | $0.716_{-.005}$ | 0.790 | 0.969 | 0.727 | $0.601_{+.012}$ | 0.725 |
| $\alpha=0.1$ | FedAvg | 0.898 | 0.729 | 0.790 | 0.787 | 0.892 | 0.574 | 0.623 | 0.657 | 0.940 | 0.533 | 0.440 | 0.577 |
| | FedAvg+ | 0.880 | 0.744 | $0.808_{+.018}$ | 0.797 | 0.878 | 0.595 | $0.647_{+.024}$ | 0.672 | 0.958 | 0.530 | $0.485_{+.045}$ | 0.598 |
| | FedProx | 0.894 | 0.715 | 0.792 | 0.782 | 0.893 | 0.572 | 0.618 | 0.655 | 0.948 | 0.534 | 0.448 | 0.582 |
| | FedProx+ | 0.898 | 0.736 | $0.817_{+.025}$ | 0.801 | 0.899 | 0.582 | $0.629_{+.011}$ | 0.664 | 0.943 | 0.555 | $0.469_{+.021}$ | 0.598 |
| | CReFF | 0.825 | 0.704 | 0.768 | 0.754 | 0.827 | 0.465 | 0.602 | 0.592 | 0.909 | 0.479 | 0.521 | 0.582 |
| | CReFF+ | 0.816 | 0.702 | $0.783_{+.015}$ | 0.757 | 0.835 | 0.464 | $0.623_{+.021}$ | 0.602 | 0.905 | 0.484 | $0.537_{+.016}$ | 0.589 |
| | GCL-FL | 0.840 | 0.723 | 0.761 | 0.762 | 0.915 | 0.579 | 0.520 | 0.623 | 0.902 | 0.407 | 0.331 | 0.476 |
| | GCL-FL+ | 0.810 | 0.729 | $0.814_{+.053}$ | 0.779 | 0.895 | 0.566 | $0.536_{-.016}$ | 0.620 | 0.919 | 0.430 | $0.364_{+.033}$ | 0.502 |
| | Fed-GraB | 0.855 | 0.737 | 0.816 | 0.792 | 0.871 | 0.560 | 0.627 | 0.649 | 0.902 | 0.481 | 0.534 | 0.587 |
| | Fed-GraB+ | 0.847 | 0.724 | 0.843$_{+.027}$ | 0.796 | 0.862 | 0.545 | $\mathbf{0.698}_{+.071}$ | 0.670 | 0.919 | 0.550 | $0.555_{+.021}$ | 0.626 |
| | CRePA | 0.886 | 0.754 | 0.842 | 0.816 | 0.904 | 0.599 | 0.652 | 0.681 | 0.899 | 0.582 | 0.584 | 0.646 |
| | CRePA+ | 0.885 | 0.728 | $\mathbf{0.853}_{+.011}$ | 0.809 | 0.882 | 0.612 | 0.691$_{+.039}$ | 0.698 | 0.911 | 0.598 | $\mathbf{0.603}_{+.019}$ | 0.662 |

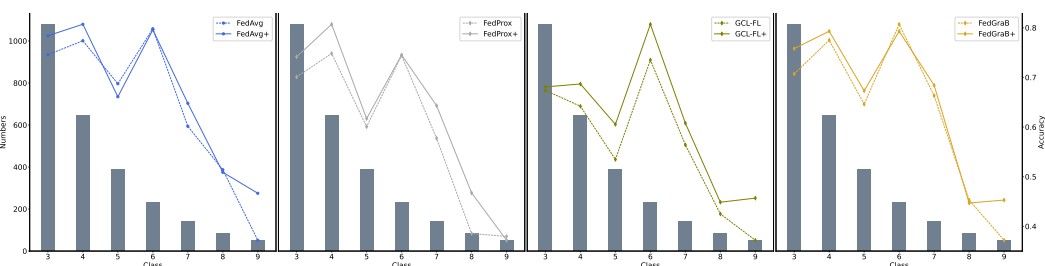

Figure 7: Comparison of performance in different categories after adding intervention model to different baselines.

**FedLT-CI's Performance:** Table 1 shows that applying FedLT-CI to different baselines can further enhance the algorithm's performance on tail data. It is worth noting that this does not impair the performance of the head and middle. Especially in the case of $\alpha$=0.5 and $IF_G$=100, where data heterogeneity and tail data are incredibly severe, the performance of intervention strategies is more significant. The performance of FedLT-CI is more significant in the case of high data heterogeneity and severe tail data distribution ($\alpha$=0.5, $IF_G$=100). For instance, adding the intervention strategy to the FedProx and GCL-FL increased the test accuracy for tail data by 4.8% and 3.4%, respectively.

To further analyze the performance of FedLT-CI, we present a comparison of different baselines and their added interventions in various categories in Fig. 7. Fig.7 shows that FedLT-CI reduces communication costs and further enhances the performance of the baseline in tail data. This further demonstrates the general effectiveness of FedLT-CI among different algorithms. We will provide comparative experimental results on the **CIFAR-100-LT** dataset in the Appendix C. For experiments on the real-world medical dataset **PTB-XL**, please refer to Appendix E.

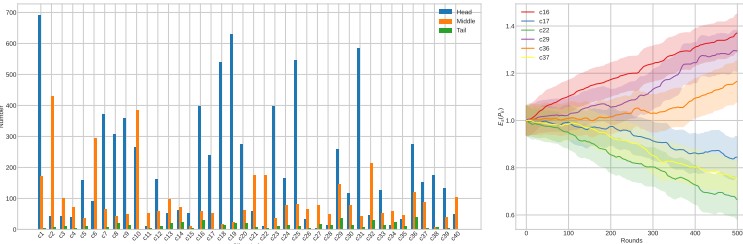

Figure 8: Client data distribution and the change in mean and variance of the client's prior distribution with communication rounds (Conducting experiments on the CIFAR-10-LT with the settings of $IF_G$=100, $\alpha$=0.5, $K$=40).

### 5.3 Model Analysis and Ablation Experiments

**CRePA model analysis:** The left graph in Fig. 8 illustrates the data sample distribution among 40 clients, while the right graph displays the changes in the mean and variance of the prior distribution for selected clients with increasing communication rounds. An analysis was conducted for six clients, including c16, c29, c36 with more tail data, and c17, c22, c36 with fewer tail data. Experimental results indicate that CRePA, by leveraging both tail and non-tail gradient information during communication, dynamically balances tail and non-tail performance as the global model updates. When the communication rounds exceed 100, CRePA begins to increase the distribution expectation for clients with a higher proportion of tail data, and this trend becomes more pronounced with increasing rounds. For example, as the model has already converged on non-tail classes during the 100 rounds, and client c37 (yellow) contains very few tail data, leading to a decrease in its prior mean during later aggregation processes. In contrast, client c16, with abundant tail data, experiences an increase in its mean. This also indicates that with increased communication rounds, the model has converged on the categories of the head and middle, while the tail data can still achieve further improvement.

**Impact of $\mathcal{F}$ on the FedLT-CI:** To investigate the effect of the hyper-parameter $\mathcal{F}$, we observed the variation in test accuracy for different values of $\mathcal{F}$, as shown in Fig. 6(a). The results indicate that regardless of the value of $\mathcal{F}$, FedLT-CI can improve the performance on tail data in imbalanced datasets. However, the smaller the $\mathcal{F}$, the relatively better the performance. Due to the low $\mathcal{F}$, clients with lower causal effects on the global model will be excluded more frequently during the aggregation process. Further analysis of $\mathcal{F}$ on other baselines will be provided in the Appendix D.1.

**Impact of $\mathcal{N}$ on the FedLT-CI:** $\mathcal{N}$ as a critical hyper-parameter of the FedLT-CI. To test its impact on tail performance, we conducted experiments on the CIFAR-10-LT dataset with $\alpha$=0.5 and $IF_G$=100. We plotted the changes in tail accuracy for different values of $\mathcal{N}$, as shown in Fig. 6(b). It is evident that there are many suitable choices for $\mathcal{N}$. Therefore, tuning $\mathcal{N}$ in causal intervention is relatively easy in this setting. For more ablation experiments, please refer to the Appendix D.2.

## 6 Conclusion

This paper proposes two innovative algorithms to tackle the challenges of data heterogeneity and the long-tail phenomenon in Fed-LT. Firstly, we introduce the CRePA, which divides gradient information into tail and non tail information to achieve online learning of weight prior distribution parameters for each client. Experimental results demonstrate that, compared to other baselines, the CRePA algorithm not only preserves the performance of non-tail data but also enhances the performance of tail data. To address the interpretability challenges in Fed-LT, we draw inspiration from Pearl's causal structure model and present the FedLT-CI. FedLT-CI assesses the causal effects of individual clients on the global model's performance on tail data and intervenes during the aggregation process to significantly enhance tail effects. Experimental results showcase that, across multiple experiments, the FedLT-CI algorithm significantly improves the performance of various baseline models on tail data. This algorithm not only achieves promising results in addressing the challenges of Fed-LT but also makes a breakthrough in interpretability. It introduces fresh ideas and methods for this direction, establishing a solid foundation for future research.

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

## A  PROPOSE FEDLT-CI METHOD DETAILS.

### A.1  METHODOLOGY DETAILS

**FedLT-CT Objective Function:** The objective function in FedLT-CT is shown in Equation 10. Due to the unknown causal effects of clients on the global aggregated model, the approximation posterior $q(c_1, c_2, \cdots, c_K | Y, Y^{'})$ is challenging. Therefore, we need to transform Equation 10 to make the approximate posterior conditional only on the gradient information $Y^{'}$.

$$Equation\ (10) = \max_{p}\max_{q} \mathbb{E}_{p^*(Y,Y')} \left[ \underbrace{\mathbb{E}_{q(c_1,c_2,\cdots,c_K|Y,Y')} \left[ \log \frac{p(c_1,c_2,\cdots,c_K,Y,Y')}{q(c_1,c_2,\cdots,c_K|Y,Y')} \right]}_{A} \right] \quad (12)$$

We will derive $A$ as follows:

$$
\begin{aligned}
A &= \int \left[ \log \frac{p(c_1, c_2, \cdots, c_K, Y, Y')}{q(c_1, c_2, \cdots, c_K | Y, Y')} \right] q(c_1, c_2, \cdots, c_K | Y, Y') dc_1 dc_2 \cdots dc_K \\
&= \int \left[ \log \frac{p(c_1, c_2, \cdots, c_K, Y, Y') q(Y)}{q(c_1, c_2, \cdots, c_K, Y | Y')} \right] q(c_1, c_2, \cdots, c_K | Y, Y') dc_1 dc_2 \cdots dc_K \\
&= \int \left[ \log \frac{p(c_1, c_2, \cdots, c_K, Y, Y')}{q(c_1, c_2, \cdots, c_K, Y | Y')} \right] \frac{q(c_1, c_2, \cdots, c_K, Y | Y')}{q(Y)} dc_1 dc_2 \cdots dc_K \\
&\quad + \int \left[ \log q(Y) \right] q(c_1, c_2, \cdots, c_K | YY') dc_1 dc_2 \cdots dc_K \\
&= \int \left[ \log \frac{p(c_1, c_2, \cdots, c_K, Y, Y')}{q(c_1, c_2, \cdots, c_K, Y | Y')} \right] \frac{q(c_1, c_2, \cdots, c_K, Y | Y')}{q(Y)} dc_1 dc_2 \cdots dc_K + \log q(Y)
\end{aligned}
\tag{13}
$$

So, we obtain an equivalent simplification of Equation 12:

$$
\begin{aligned}
\max_{p\,q} &\ \mathbb{E}_{p^*(Y,Y')} \int \left[ \log \frac{p\left(c_1, c_2, \cdots, c_K, Y, Y'\right)}{q(c_1, c_2, \cdots, c_K, Y | Y')} \right] \frac{q(c_1, c_2, \cdots, c_K, Y | Y')}{q(Y)} dc_1 dc_2 \cdots dc_K \\
&+ \log q(Y) \\
= \max_{p\,q} &\int \left[ \log \frac{p(c_1, c_2, \cdots, Y, Y')}{q(c_1, c_2, \cdots, Y | Y')} \right] \frac{q(c_1, c_2, \cdots, c_K, Y | Y')}{q(Y)} p^*(YY') dc_1 dc_2 \cdots dc_K dY dY' \\
&+ \int p^*(YY') \log q(Y) dY dY' \\
= \max_{p\,q} &\int \underbrace{ \mathbb{E}_{q(c_1, c_2, \cdots, c_K, Y | Y')} \frac{p^*(Y)}{q(Y)} \left[ \log \frac{p(c_1, c_2, \cdots, Y, Y')}{q(c_1, c_2, \cdots, Y | Y')} \right] p^*(Y') dY'}_{B} + \mathbb{E}_{p^*(YY')} \log q(Y)
\end{aligned}
\tag{14}
$$

Then, we transformed $B$ into:

$$
\begin{aligned}
B &= \mathbb{E}_{p^*(Y')} \left[ \mathbb{E}_{q(c_1, c_2, \cdots, c_K, Y | Y')} \frac{p^*(Y)}{q(Y)} \left[ \log \frac{p(c_1, c_2, \cdots, Y, Y')}{q(c_1, c_2, \cdots, Y | Y')} \right] \right] \\
&= \mathbb{E}_{p^*(Y')} \left[ \int \frac{p^*(Y)}{q(Y)} \left[ \log \frac{p(c_1, c_2, \cdots, Y, Y')}{q(c_1, c_2, \cdots, Y | Y')} \right] q(c_1, c_2, \cdots, c_K, Y | Y') dc_1 dc_2 \cdots dc_K dY \right] \\
&= \mathbb{E}_{p^*(Y')} \left[ \int \left[ \log \frac{p(c_1, c_2, \cdots, c_K, Y, Y')}{q(c_1, c_2, \cdots, c_K, Y | Y')} \right] \frac{q(Y | c_1, c_2, \cdots, c_K, Y') q(c_1, c_2, \cdots, c_K | Y')}{q(Y)} \right. \\
&\quad \left. + p^*(Y) dY dc_1 dc_2 \cdots dc_K \right] \\
&= \mathbb{E}_{p^*(YY')} \left[ \mathbb{E}_{q(c_1, c_2, \cdots, c_K | Y')} \frac{q(Y | c_1, c_2, \cdots, c_K, Y')}{q(Y)} \log \frac{p(c_1, c_2, \cdots, c_K, Y, Y')}{q(c_1, c_2, \cdots, c_K, Y | Y')} \right]
\end{aligned}
\tag{15}
$$

$$\max_{p \; q} \mathbb{E}_{p^*(YY')} \left[ \log q(Y) + \mathbb{E}_{q(c_1, c_2, \cdots, c_K | Y')} \frac{q(Y | c_1, c_2, \cdots, c_K, Y')}{q(Y)} \log \frac{p(c_1, c_2, \cdots, c_K, Y, Y')}{q(c_1, c_2, \cdots, c_K, Y | Y')} \right]$$

$$= \max_{p \; q} \mathbb{E}_{p^*(YY')} \left[ \log q(Y) + \mathbb{E}_{q(c_1, c_2, \cdots, c_K | Y')} \frac{q(Y | c_1, c_2, d \cdots, c_K, Y')}{q(Y)} \right.$$

$$\left. \log \frac{p(Y | c_1, c_2, \cdots, c_K, Y') p(Y', c_1, c_2, \cdots, c_K)}{q(Y | c_1, c_2, \cdots, c_K, Y') q(c_1, c_2, \cdots, c_K | Y')} \right]$$

$$= \max_{p \; q} \mathbb{E}_{p^*(YY')} \left[ \log q(Y) + \mathbb{E}_{q(c_1, c_2, \cdots, c_K | Y')} \frac{q(Y | c_1, c_2, d \cdots, c_K, Y')}{q(Y)} \log \frac{p(Y', c_1, c_2, \cdots, c_K)}{q(c_1, c_2, \cdots, c_K | Y')} \right]$$

$$= \max_{p \; q} \mathbb{E}_{p^*(YY')} \left[ \log q(Y) + \mathbb{E}_{q(c_1, c_2, \cdots, c_K | Y')} \frac{q(Y | c_1, c_2, d \cdots, c_K, Y')}{q(Y)} \right.$$

$$\left. \log \frac{p(Y' | c_1, c_2, \cdots, c_K) p(c_1) p(c_2) \cdots p(c_K)}{q(c_1 | Y') q(c_2 | Y') \cdots q(c_K | Y')} \right] \tag{16}$$

$q(Y)$ can be written as:

$$q(Y) = \int q(Y | c_1, c_2, \cdots, c_K) q(c_1, c_2, \cdots, c_K | Y')$$
$$= p(Y | do(c_i)) \tag{17}$$
$$= \mathbb{E}_{p(c_1 | Y'), \cdots, p(c_K | Y')} [p(Y | c_1, \cdots, c_i \cdots, c_K)]$$

Where $p(Y | do(c_i))$ represents the causal effect of $c_i$ on $Y$ and can be computed through sampling. Therefore, we also obtain the objective loss function of FedLT-CI (i.e., Equation 11).

### A.2 THE COMMUNICATION COST ANALYSIS

Proof of Reduced Communication Cost:

Assuming the size of the global model parameters is $S_g$, with the size of the parameters in the classification layer denoted as $S_{g_c}$, it is evident that $S_g >> S_{g_c}$. Let $K$ be the number of clients.

Firstly, in the absence of causal intervention, let the transmission cost for $T$ communication rounds be denoted as $Cost$. Clearly, $Cost = 2TKS_g$.

When causal intervention strategy is introduced, with $\mathcal{F}$ as the intervention frequency and $\mathcal{N}$ as the number of intervened clients, let $Cost_{int}$ represent the communication cost at this point. We get:

$$Cost_{int} = \left( T - \left\lfloor \frac{T}{\mathcal{F}+1} \right\rfloor \right) (S_g + S_{g_c}) K + \left\lfloor \frac{T}{\mathcal{F}+1} \right\rfloor [S_g (K - \mathcal{N}) + K S_{g_c}] + TKS_g \tag{18}$$

We define $\gamma = \left\lfloor \frac{T}{\mathcal{F}+1} \right\rfloor$. The we obtain:

$$Cost_{int} = (T - \gamma)(S_g + S_{g_c}) K + \gamma [S_g (K - \mathcal{N}) + K S_{g_c}] + TKS_g$$
$$= TKS_g + TKS_{g_c} - \gamma S_g N + TKS_g \tag{19}$$
$$= (2TK - QN)S_g + TKS_{g_c}$$

Finally, we calculate $\frac{Cost_{int}}{Cost}$:

$$\frac{Cost_{int}}{Cost} = \frac{(2TK - QN)S_g + TKS_{g_c}}{2TKS_g}$$
$$= 1 - \frac{QN}{2TK} + \frac{S_{g_c}}{2S_g} \tag{20}$$
$$\stackrel{(b_1)}{=} 1 - \frac{QN}{2TK} = 1 - \frac{N}{2TK} \left\lfloor \frac{T}{\mathcal{F}+1} \right\rfloor$$

$b_1$ is $S_g >> S_{g_c}$.

From the conclusions, we know that $\frac{N}{2TK} \left\lfloor \frac{T}{\mathcal{F}+1} \right\rfloor$ represents the percentage of communication cost reduction, and it is correlated with the intervention frequency $\mathcal{F}$ and the number of intervened clients $\mathcal{N}$. The communication cost decreases with an increase in the number of $\mathcal{N}$ (i.e., the larger the number of $\mathcal{N}$, the smaller the communication cost, but one must also consider the impact on performance). The communication cost increases with an increase in the intervention frequency $\mathcal{F}$ (i.e., the more frequent the intervention, the smaller the communication cost). When $F = 0$ (i.e., intervention in every aggregation), according to the conclusion, the percentage of communication cost reduction is $\frac{N}{2K}$. For example, if $N$ is set to $\frac{1}{4K}$ (as in most of our experiments), the communication cost can be reduced by $1/8$.

## B ALGORITHMS

### B.1 CRePA ALGORITHM

---

**Algorithm 1** CRePA: Client Re-weighted Prior Analyzer. $\theta_t$ denotes the parameters of the global model at the $t$-th round, and $w_{pk}$ denotes the prior distribution parameters for client $k$. $G_{g,nott}$ and $G_{g,tail}$ denote the non-tail gradient and tail gradient change information of the global model, respectively.

---

**Server executes:**
  Initialize $\theta_0, w_{p1}, w_{p2}, \cdots, w_{pK}, G_{g,nott} = \mathbf{0}, G_{g,tail} = \mathbf{0}$
  **for** each round $t = 1, \cdots, T$ **do**
    **for** each client $k \in [K]$ **in parallel do**
      $\theta_{t+1}, G_{k,nott}, G_{k,tail} \leftarrow$ ClientUpdate$(k, \theta_t)$
    **end for**
    $\mathcal{B} \leftarrow$ Obtain set $\mathcal{B}$ through Eq. (4)
    $\mathcal{B}' \leftarrow$ Sort the set $\mathcal{B}$ in descending order
    **for** each client $k \in [K]$ **do**
      $w_{pk} \leftarrow w_{pk} - \eta \nabla \mathcal{L}_{prior}$ // by Eq. (5)
    **end for**
    $\theta_{t+1} = \sum_{k=1}^{K} \frac{\mathbb{E}_s(P_k)}{\sum_{k=1}^{K} \mathbb{E}_s(P_k)} \theta_{k,t+1}$ // Calculate $E$ through Eq. (3)
    $G_{g,nott} \leftarrow \sum_{k=1}^{K} \frac{\mathbb{E}_s(P_k)}{\sum_{k=1}^{K} \mathbb{E}_s(P_k)} G_{k,nott}$ ,
    $G_{g,tail} \leftarrow \sum_{k=1}^{K} \frac{\mathbb{E}_s(P_k)}{\sum_{k=1}^{K} \mathbb{E}_s(P_k)} G_{k,tail}$
  **end for**

**ClientUpdate**$(k, \theta)$: // Run on client k
  // Cache the change vectors of tail and non-tail gradients from this iteration.
  Initialize $G_{k,nott} = \mathbf{0}, G_{k,tail} = \mathbf{0}$
  $B \leftarrow$ (split $D^k$ into batches of size $B$)
  // $E$ denotes the number of local training epochs.
  **for** each local epoch $i$ from 1 to $E$ **do**
    **for** batch $b \in B$ **do**
      $\theta \leftarrow \theta - \eta_k \nabla \mathcal{L}(\theta; b)$
      $G_{k,nott} \leftarrow G_{k,nott} + \eta_k \nabla_{W_{k,nott}}(\mathcal{L})$
      $G_{k,tail} \leftarrow G_{k,tail} + \eta_k \nabla_{W_{k,tail}}(\mathcal{L})$
    **end for**
    return $\theta, G_{k,nott}, G_{k,tail}$ to server
  **end for**

---

### B.2 FEDLT-CI ALGORITHM

**Algorithm 2** FedLT-CI: Federated Long-Tail Causal Intervention Model. As an example, we apply the FedLT-CI algorithm to the FedAvg baseline. $\theta_{CI}$ denotes the parameters of the FedLT-CI model.

---

**Server executes:**
  Initialize $\theta_0, \theta_{CI}, G_{g,tail} = \mathbf{0}$.
  **for** each round $t = 1, \cdots, T$ **do**
    **for** each client $k \in [K]$ **in parallel do**
      $\theta_{k,t+1}, G_{k,tail} \leftarrow \text{ClientUpdate}(k, \theta_t)$
    **end for**
    $Y^{'} \leftarrow$ Using Eq. (7), compute $Y^{'}$ through the $G_{k,tail}$ and $G_{g,tail}$
    // $\mathcal{L}_{CI}$ denotes the loss function of the causal model (as shown in Eq. 11)
    $\theta_{CI} \leftarrow \theta_{CI} - \eta \mathcal{L}_{CI}$
    **for** each client $k \in [k]$ **do**
      $\tau_k \leftarrow$ compute the causal effect of client $k$ using Eq. 6
    **end for**
    $S_t \leftarrow$ select $\mathcal{N}$ clients with smaller $\tau$ values
    // $n_k$ denotes the dataset size of client $k$
    $n = \sum_{k \notin S_t} n_k$ // $n = \sum_{k=1}^{K} n_k$ is the original FedAvg model process
    $\theta_{t+1} = \sum_{k \notin S_t} \frac{n_k}{n} \theta_{k,t+1}$ // $\theta_{t+1} = \sum_{k=1}^{K} \frac{n_k}{n} \theta_{k,t+1}$ is the original FedAvg model process
  **end for**

**ClientUpdate$(k, \theta)$:** // Run on client k
  $B \leftarrow$ (split $D^k$ into batches of size $B$)
  Initialize $G_{k,tail} = \mathbf{0}$
  // $E$ denotes the number of local training epochs.
  **for** each local epoch $i$ from 1 to $E$ **do**
    **for** batch $b \in B$ **do**
      $\theta \leftarrow \theta - \eta_k \nabla \mathcal{L}(\theta; b)$
      $G_{k,tail} \leftarrow G_{k,tail} + \eta_k \nabla_{W_{k,tail}}(\mathcal{L})$
    **end for**
    return $\theta, G_{k,tail}$ to server
  **end for**

---

## C  ADDITIONAL RESULTS: COMPARISON WITH STATE-OF-THE-ART METHODS

**Evaluation on CIFAR-10-LT.** The experiments in this section supplement the evaluation of CIFAR-10-LT presented in Section 5.2. Fig. 9, further illustrating the performance of the tail during the entire communication iteration process when the causal intervention strategy is applied to different models. Fig. 9 shows that the proposed causal model can quickly understand the causal mechanism and variables through online learning, leading to an improvement in tail performance.

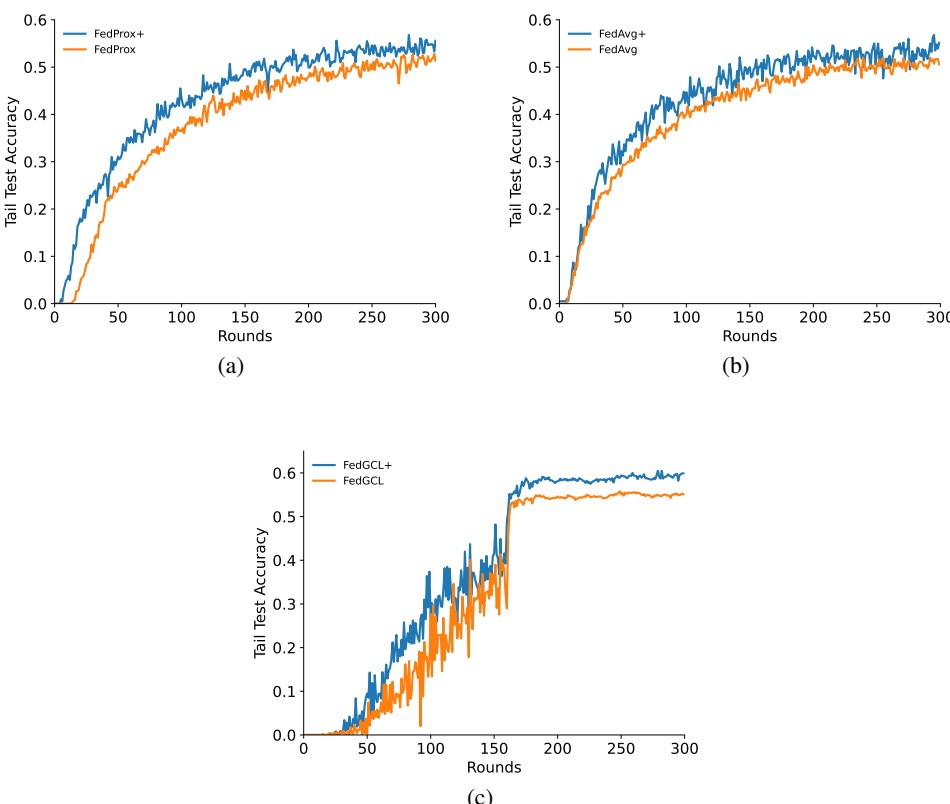

Figure 9: Comparison of the effects on the tail data for various methods with and without our causal intervention on the CIFAR-10-LT dataset ($IF_G$=100, $\alpha$=0.5).

**How is the comprehensive impact of FedLT-CI assessed across the entire iteration process?** Based on the intervention status of the Fed-LT model during the training process, clients are excluded. This experiment provides a comprehensive evaluation of the proposed intervention model, and we employ the following evaluation method: during the process of 300 communication rounds, with an intervention frequency set at $F = 2$ and the number of intervened clients at $\mathcal{N} = 1/4K$ ($K$=40), we collected statistics on the clients that were intervened and excluded in rounds 100 to 300. We selected the top $\mathcal{N}$ clients with the highest intervention exclusion frequencies. Subsequently, we conducted comparative experiments to compare the model performance between aggregating with exclusion of these $\mathcal{N}$ clients and training with all clients. Specifically, in the case of $K = 40$, we permanently excluded the $\mathcal{N}$ clients that were removed and compared the results with the model trained with all clients, as shown in Table 2. From an imprecise perspective, by analyzing the comparison in Table 2, we can comprehensively assess how the proposed intervention model functions throughout the entire iteration process. It is evident from Table 2 that FedLT-CI is reliable in discerning the causal effects of clients on the aggregation model, particularly in FedAvg, FedProx, and GCL-GraB models, where FedLT-CI leads to an improvement in tail class accuracy by 2.2%, 1.3%,

and 2.2%, respectively. This indicates that the introduction of FedLT-CI significantly enhances the model's performance on tail data.

Table 2: Comparing the effects of selecting $\mathcal{N}$ clients with low causal effects during the intervention process of the Fed-LTCI model for permanent exclusion from aggregation.

| METHOD | $IF_G = 100$ | | | |
|---|---|---|---|---|
| | MANY | MED | FEW | ALL |
| FEDAVG | 0.971 | 0.706 | 0.523 | 0.686 |
| FEDAVG+ | 0.975 | 0.658 | $0.545_{+.022}$ | 0.676 |
| FEDPROX | 0.973 | 0.700 | 0.527 | 0.686 |
| FEDPROX+ | 0.974 | 0.652 | $0.540_{+.013}$ | 0.672 |
| FED-GRAB | 0.964 | 0.716 | 0.581 | 0.712 |
| FED-GRAB+ | 0.963 | 0.695 | $0.603_{+.022}$ | 0.712 |

**Evaluation on CIFAR-100-LT.**

Table 3: Top 1 test accuracies of our methods and SOTA methods on CIFAR-100-LT with diverse imbalanced data settings. + indicates the incorporation of our intervention algorithm in the baseline aggregation task.

| METHOD | $IF_G = 10$ | | | | $IF_G = 50$ | | | | $IF_G = 100$ | | | |
|---|---|---|---|---|---|---|---|---|---|---|---|---|
| | MANY | MED | FEW | ALL | MANY | MED | FEW | ALL | MANY | MED | FEW | ALL |
| FEDAVG | 0.699 | 0.594 | 0.423 | 0.547 | 0.672 | 0.481 | 0.196 | 0.405 | 0.674 | 0.449 | 0.133 | 0.367 |
| FEDAVG+ | 0.695 | 0.614 | $0.439_{+.0.016}$ | 0.560 | 0.682 | 0.493 | $0.209_{+0.013}$ | 0.417 | 0.673 | 0.443 | $0.142_{+0.009}$ | 0.369 |
| FEDPROX | 0.704 | 0.589 | 0.423 | 0.546 | 0.686 | 0.493 | 0.202 | 0.415 | 0.669 | 0.429 | 0.125 | 0.355 |
| FEDPROX+ | 0.709 | 0.606 | $0.436_{+0.013}$ | 0.559 | 0.688 | 0.486 | $0.211_{+0.009}$ | 0.416 | 0.664 | 0.440 | $0.137_{+.012}$ | 0.364 |
| CREFF | 0.685 | 0.568 | 0.397 | 0.523 | 0.641 | 0.435 | 0.202 | 0.383 | 0.634 | 0.398 | 0.132 | 0.339 |
| CREFF+ | 0.691 | 0.571 | $0.413_{+0.016}$ | 0.532 | 0.636 | 0.426 | $0.227_{+0.025}$ | 0.388 | 0.639 | 0.406 | $0.136_{+0.04}$ | 0.345 |
| GCL-FL | 0.681 | 0.587 | 0.416 | 0.537 | 0.636 | 0.491 | 0.209 | 0.407 | 0.591 | 0.406 | 0.138 | 0.336 |
| GCL-FL+ | 0.677 | 0.595 | $0.421_{+0.005}$ | 0.542 | 0.619 | 0.498 | $0.205_{-0.004}$ | 0.405 | 0.584 | 0.409 | $0.144_{+0.06}$ | 0.338 |
| FED-GRAB | 0.705 | 0.616 | 0.461 | 0.572 | 0.717 | 0.563 | 0.247 | 0.467 | 0.707 | 0.539 | 0.159 | 0.421 |
| FED-GRAB+ | 0.701 | 0.638 | $\mathbf{0.470}_{+.009}$ | 0.583 | 0.727 | 0.573 | $\mathbf{0.263}_{+.016}$ | 0.480 | 0.716 | 0.544 | $\mathbf{0.168}_{+.009}$ | 0.428 |
| CREPA | 0.703 | 0.601 | 0.439 | 0.557 | 0.709 | 0.524 | 0.229 | 0.443 | 0.674 | 0.459 | 0.142 | 0.375 |
| CREPA+ | 0.727 | 0.627 | $0.459_{+.020}$ | 0.580 | 0.706 | 0.527 | $0.243_{+.014}$ | 0.449 | 0.693 | 0.476 | $0.157_{+.015}$ | 0.392 |

**CRePA's Performance.** In Table 3, we summarize the experimental results of all methods under a Non-IID setting ($\alpha = 0.5$) and three different IF settings. In the context of the extremely imbalanced CIFAR-100-LT dataset, CRePA exhibits superior tail testing accuracy over other baseline methods in most scenarios. For instance, under the $IF_G=10$ setting, the tail accuracy is improved by 1.6%, 1.6%, and 2.3% compared to FedAvg, FedProx, and GCL-FL, respectively. Similarly, under the $IF_G=50$ setting, the improvements are 3.3%, 2.7%, and 2% compared to FedAvg, FedProx, and GCL-FL, respectively. Meanwhile, in most scenarios, CRePA also achieves superior overall performance. For example, under the $IF_G=50$ condition, the overall accuracy improvements compared to FedAvg, FedProx, and GCL-FL are 3.8%, 2.8%, and 3.6%, respectively.

The reason for this superiority is that when the model is tasked with a classification involving a large number of categories, it typically converges quickly on non-tail data. In such cases, CRePA leverages the gradient variation information of the tail data from each client during communication, thereby enhancing the weighted prior distribution expectations for clients with relatively rich tail data. This, in turn, further improves performance on tail data during subsequent server-side model aggregation.

**FedLT-CI's Performance:** In Table 3, the introduction of the intervention algorithm significantly enhances the performance of different algorithms, especially in terms of tail data. Observing the data in the table, especially in the case of high data heterogeneity and severe tail data (i.e., $\alpha$=0.5, $IF_G$=50), the performance of the intervention strategy is particularly prominent. For example, adding the intervention strategy in FedAvg and Fed-GraB algorithms can respectively improve the testing accuracy in tail data by 1.3% and 1.6%.

The additional experiments presented here, as well as those conducted in Section 5.2, thoroughly validate the strong performance of CRePA in federated long-tail learning. Furthermore, the introduced causal intervention model can further enhance the performance of FL algorithms on tail data.

**On the CIFAR-100-LT, a comparison of test accuracy for the last ten classes:** Table 4 presents a performance comparison of various baselines on the last ten tail classes of the CIFAR-100-LT dataset. By observing the AVG. column, the following conclusions can be drawn: (i) CRePA achieves the best performance without intervention (2.3%, 2.1%, and 2.1% higher than FedAvg, FedProx, and CReFF, respectively). (ii) Our proposed intervention strategy significantly improves the model's performance on tail classes. The table shows that introducing the intervention strategy in different baselines enhances of tail class performance. For instance, GCL-FL exhibits a 1.1% improvement and achieves the best performance among all results.

Table 4: Comparison of the highest average test accuracy on the last 10 classes of tail data on CIFAR-100-LT ($\alpha = 0.5, IF_G = 100$). Bold: best results.

| METHOD | | | | | $IF_G = 100$ | | | | | | |
|---|---|---|---|---|---|---|---|---|---|---|---|
| | CLASS91 | CLASS92 | CLASS93 | CLASS94 | CLASS95 | CLASS96 | CLASS97 | CLASS98 | CLASS99 | CLASS100 | AVG. |
| FEDAVG | 0.00 | 0.02 | 0.01 | 0.00 | 0.04 | 0.04 | 0.00 | 0.00 | 0.00 | 0.03 | 0.014 |
| FEDAVG+ | 0.01 | 0.07 | 0.05 | 0.02 | 0.08 | 0.02 | 0.00 | 0.00 | 0.01 | 0.02 | $0.028_{+.014}$ |
| FEDPROX | 0.00 | 0.01 | 0.03 | 0.00 | 0.04 | 0.03 | 0.00 | 0.01 | 0.01 | 0.03 | 0.016 |
| FEDPROX+ | 0.00 | 0.05 | 0.03 | 0.02 | 0.02 | 0.02 | 0.00 | 0.01 | 0.02 | 0.03 | $0.020_{+.004}$ |
| CREFF | 0.01 | 0.03 | 0.01 | 0.01 | 0.05 | 0.01 | 0.01 | 0.0 | 0.0 | 0.03 | 0.016 |
| CREFF+ | 0.02 | 0.04 | 0.03 | 0.00 | 0.07 | 0.00 | 0.01 | 0.01 | 0.0 | 0.03 | $0.021_{+.005}$ |
| GCL-FL | 0.00 | 0.06 | 0.01 | 0.02 | 0.08 | 0.08 | 0.00 | 0.03 | 0.00 | 0.03 | 0.031 |
| GCL-FL+ | 0.00 | 0.11 | 0.00 | 0.03 | 0.15 | 0.09 | 0.01 | 0.00 | 0.00 | 0.03 | $\mathbf{0.042}_{+.011}$ |
| FED-GRAB | 0.04 | 0.1 | 0.01 | 0.00 | 0.16 | 0.01 | 0.00 | 0.01 | 0.00 | 0.06 | 0.039 |
| FED-GRAB+ | 0.04 | 0.11 | 0.03 | 0.02 | 0.09 | 0.04 | 0.03 | 0.01 | 0.00 | 0.04 | $0.041_{+.002}$ |
| CREPA | 0.03 | 0.04 | 0.01 | 0.03 | 0.08 | 0.06 | 0.05 | 0.01 | 0.00 | 0.06 | 0.037 |
| CREPA+ | 0.04 | 0.05 | 0.02 | 0.02 | 0.13 | 0.06 | 0.02 | 0.02 | 0.0 | 0.06 | $\mathbf{0.042}_{+.005}$ |

Fig. 10 displays the test accuracy of various models on the last 10 tail classes of CIFAR-100-LT. The graph provides a clearer view, demonstrating the consistent effectiveness of our proposed FedLT-CI in improving the performance on tail classes.

**On the CIFAR-100-LT, set a larger number of clients (K = 40).** In contrast to the previous experiment where the client quantity $K$ was set differently, this time, the experiment sets $K$ to 40 to further observe the effect of intervention in more diverse categories. By comparing the results with and without intervention, we present the performance variation of the model in Fig. 11. The figure clearly demonstrates that our proposed causal intervention model, when applied to any baseline model, can enhance the performance on tail class data while maintaining the overall performance.

# D    ADDITIONAL RESULTS: ABLATION EXPERIMENTS

## D.1    IMPACT OF INTERVENTION FREQUENCY $\mathcal{F}$ ON THE FEDLT-CI

To further investigate the impact of the intervention frequency $\mathcal{F}$ on the algorithms, We supplement the experiments in Section 5.3 here. As shown in Fig. 12(a), we plot the curves of test accuracy over iterations for different values of $\mathcal{F}$. The results once again confirmed the findings in Section 5.3, indicating that FedLT-CI can improve the performance of other models on tail data in imbalanced datasets. Moreover, it was observed that the algorithm performed relatively better when $\mathcal{F}$ was set to smaller values. This is because, with a minor intervention frequency, clients with lower causal effects on the global model would be excluded from aggregation more frequently, leading to enhanced algorithm performance.

## D.2    IMPACT OF $\mathcal{N}$ ON THE FEDLT-CI

Here, we supplement the experiments investigating the impact of $\mathcal{N}$ on intervention performance from Section 5.3. As shown in Fig. 12(b), we plot the curve of test accuracy over iterations for different values of $\mathcal{N}$. The experimental results are consistent with the conclusions from Section 5.3, indicating that various values of $\mathcal{N}$ can improve the model's performance on the tail, highlighting the ease of tuning the parameter $\mathcal{N}$ in causal intervention.

## D.3    STABILITY EXPERIMENT

Table 5 illustrates the frequency of achieving a certain tail testing accuracy on the two datasets. Informally, the more times a model reaches or exceeds this accuracy, the more stable the model

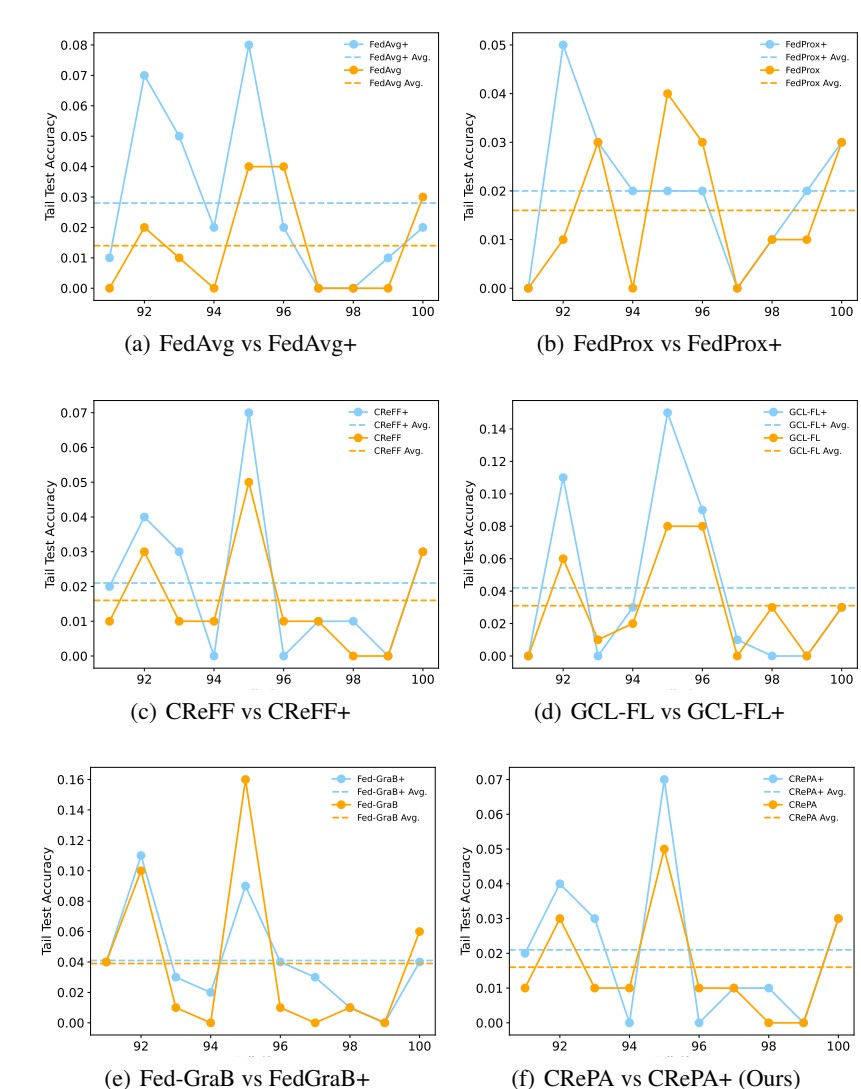

Figure 10: Performance visualization of models on the last 10 classes of tail test data on CIFAR-100-LT.

is. From this table, we find that in most scenarios, our FedLT-CI achieves a higher frequency, demonstrating that the intervention strategy enhances the performance of each baseline in tail classes and exhibits strong stability.

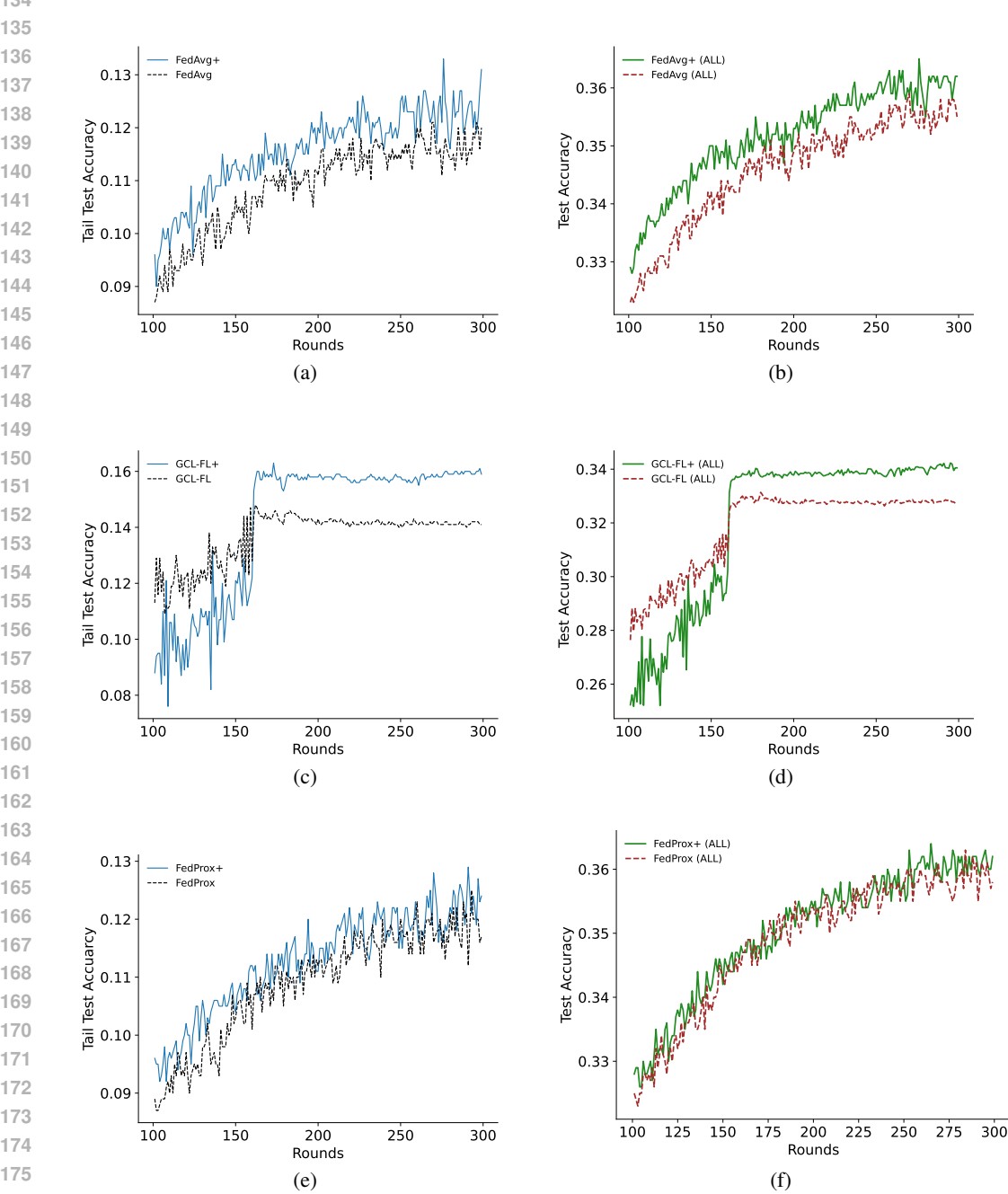

Figure 11: Comparing the intervention effects and overall performance of different models on CIFAR-100-LT with a large client number $K$.

# E  EXPERIMENTAL RESULTS FROM REAL-WORLD DATASETS

The experimental results on the PTB-XL dataset are shown in Table 6, where we compare various FL algorithms across different categories (Many, Med, Few). Without intervention, the CRePA method demonstrates the best performance on tail data (Few). For example, at $IF_G = 100$, CRePA achieves a Few category score of 0.100, significantly outperforming other non-intervention algorithms such as FedAvg (0.053) and FedGraB (0.082).

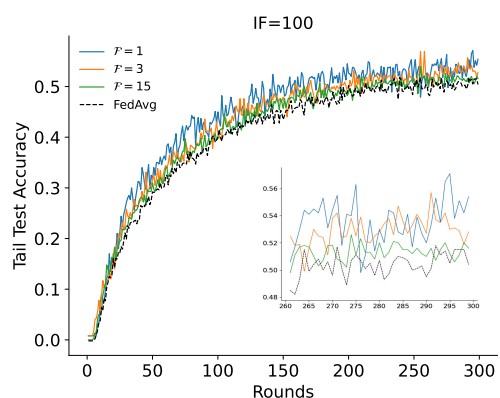

(a) Comparison of tail class performance under different $\mathcal{F}$ values.

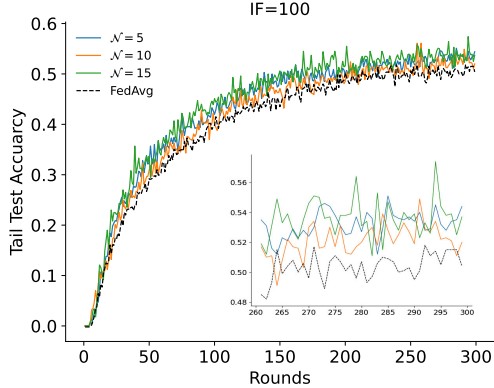

(b) Comparison of tail class performance under different $\mathcal{N}$ values.

Figure 12: Ablation experiments under FedAvg algorithm. The black dashed line represents the performance without causal intervention algorithm.

After the intervention, the performance of all algorithms improved. For example, when $IF_G = 100$, the score for FedAvg in the Few category increased from 0.053 to 0.072, and Fed-GraB's score rose from 0.082 to 0.102. Among them, CRePA+ achieved the best result. Specifically, when $IF_G = 50$, CRePA+ scored 0.144 in the Few category. Moreover, CRePA+ also demonstrated excellent performance in the overall metric (All categories), reaching 0.429 when $IF_G = 100$, showing that it not only optimized the tail performance but also maintained an improvement in the head performance.

Figure 13 shows a comparison of ROC curves for different FL algorithms before and after the intervention, across different categories. Each subplot corresponds to a comparison of algorithms, with the left side displaying head data (Head), the middle showing tail data (Tail), and the right side showing rare disease data (HYP), where HYP represents the rare disease category with the fewest samples. As seen in the figure, the intervention improves the performance of all models on the tail data. For example, FedAvg's score on the tail increased from 0.488 to 0.526, and FedGraB's score increased from 0.490 to 0.520.

Figure 14 shows a comparison of the ROC curves of different federated learning algorithms and our proposed CRePA, without any interventions, across different categories. Each subplot displays the comparison for one algorithm, with the left side showing the head data (Head) and the right side showing the tail data (Tail). As seen in the figure, CRePA achieves the best performance in the tail category without sacrificing the performance on the head data.

Table 5: Comparing the frequencies of models achieving a specified accuracy or higher in the tail classes across different datasets.

| METHOD | ACCURACY | CIFAR-10-LT | ACCURACY | CIFAR-100-LT |
|---|---|---|---|---|
| FEDAVG | 0.520 | 4 | 0.125 | 74 |
| FEDAVG+ | | 38 | | 108 |
| FEDPROX | 0.520 | 9 | 0.120 | 11 |
| FEDPROX+ | | 32 | | 73 |
| CREFF | 0.530 | 18 | 0.125 | 24 |
| CREFF+ | | 79 | | 36 |
| GCL-FL | 0.550 | 45 | 0.135 | 11 |
| GCL-FL+ | | 85 | | 38 |
| FED-GRAB | 0.570 | 31 | 0.150 | 56 |
| FED-GRAB+ | | 30 | | 86 |
| CREPA | 0.570 | 38 | 0.135 | 48 |
| CREPA+ | | 47 | | 98 |

Table 6: Test accuracies of our methods and SOTA methods on real-word datasets with diverse heterogeneous data settings. + indicates the incorporation of our intervention algorithm in the baseline aggregation task. Bold: best results. Underlined: the best results without intervention.

| SETTING | METHOD | $IF_G = 50$ | | | | $IF_G = 100$ | | | |
|---|---|---|---|---|---|---|---|---|---|
| | | MANY | MED | FEW | ALL | MANY | MED | FEW | ALL |
| $\alpha = 0.1$ | FEDAVG | 0.817 | 0.219 | 0.061 | 0.399 | 0.740 | 0.185 | 0.053 | 0.359 |
| | FEDAVG+ | 0.767 | 0.158 | $0.125_{+.064}$ | 0.418 | 0.742 | 0.222 | $0.072_{+.019}$ | 0.395 |
| | FEDPROX | 0.866 | 0.148 | 0.043 | 0.423 | 0.812 | 0.227 | 0.039 | 0.411 |
| | FEDPROX+ | 0.756 | 0.133 | $0.085_{+.042}$ | 0.373 | 0.742 | 0.161 | $0.068_{+.029}$ | 0.364 |
| | CREFF | 0.818 | 0.132 | 0.083 | 0.422 | 0.845 | 0.029 | 0.070 | 0.409 |
| | CREFF+ | 0.772 | 0.138 | $0.108_{+.025}$ | 0.411 | 0.776 | 0.074 | $\mathbf{0.118}_{+.048}$ | 0.406 |
| | DISA-FL | 0.870 | 0.178 | 0.021 | 0.439 | 0.805 | 0.244 | 0.034 | 0.413 |
| | DISA-FL+ | 0.871 | 0.106 | $0.060_{+.039}$ | 0.414 | 0.798 | 0.167 | $0.081_{+.042}$ | 0.406 |
| | FED-GRAB | 0.747 | 0.253 | 0.091 | 0.412 | 0.737 | 0.095 | 0.082 | 0.362 |
| | FED-GRAB+ | 0.772 | 0.226 | $0.117_{+.026}$ | 0.428 | 0.740 | 0.123 | $0.102_{+.0.020}$ | 0.390 |
| | CREPA | 0.741 | 0.265 | $\underline{0.120}$ | 0.423 | 0.765 | 0.111 | $\underline{0.100}$ | 0.398 |
| | CREPA+ | 0.714 | 0.260 | $\mathbf{0.144}_{+.024}$ | 0.418 | 0.777 | 0.219 | $\mathbf{0.118}_{+.018}$ | 0.429 |

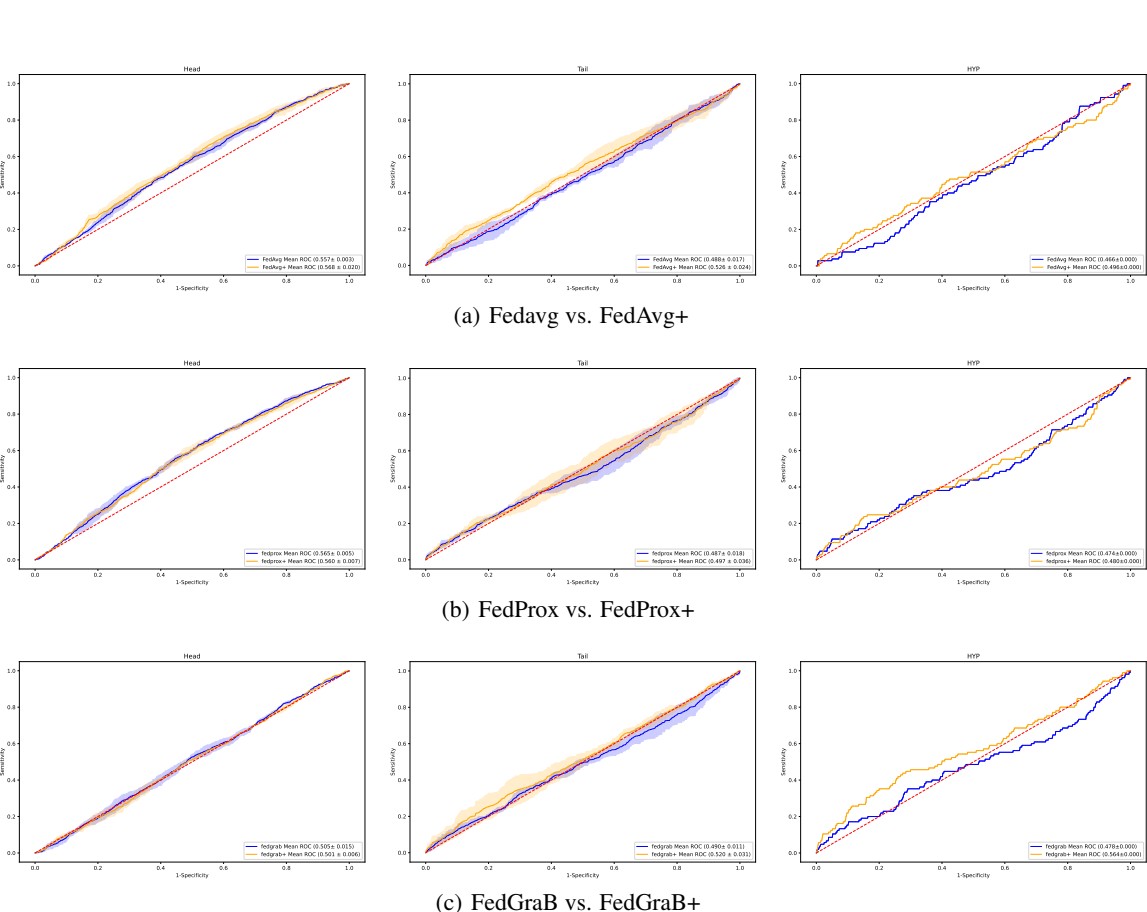

(a) Fedavg vs. FedAvg+

(b) FedProx vs. FedProx+

(c) FedGraB vs. FedGraB+

Figure 13: Comparison of ROC curves after intervention with different FL models. Among them, HYP represents the class of rare diseases with the smallest sample size.

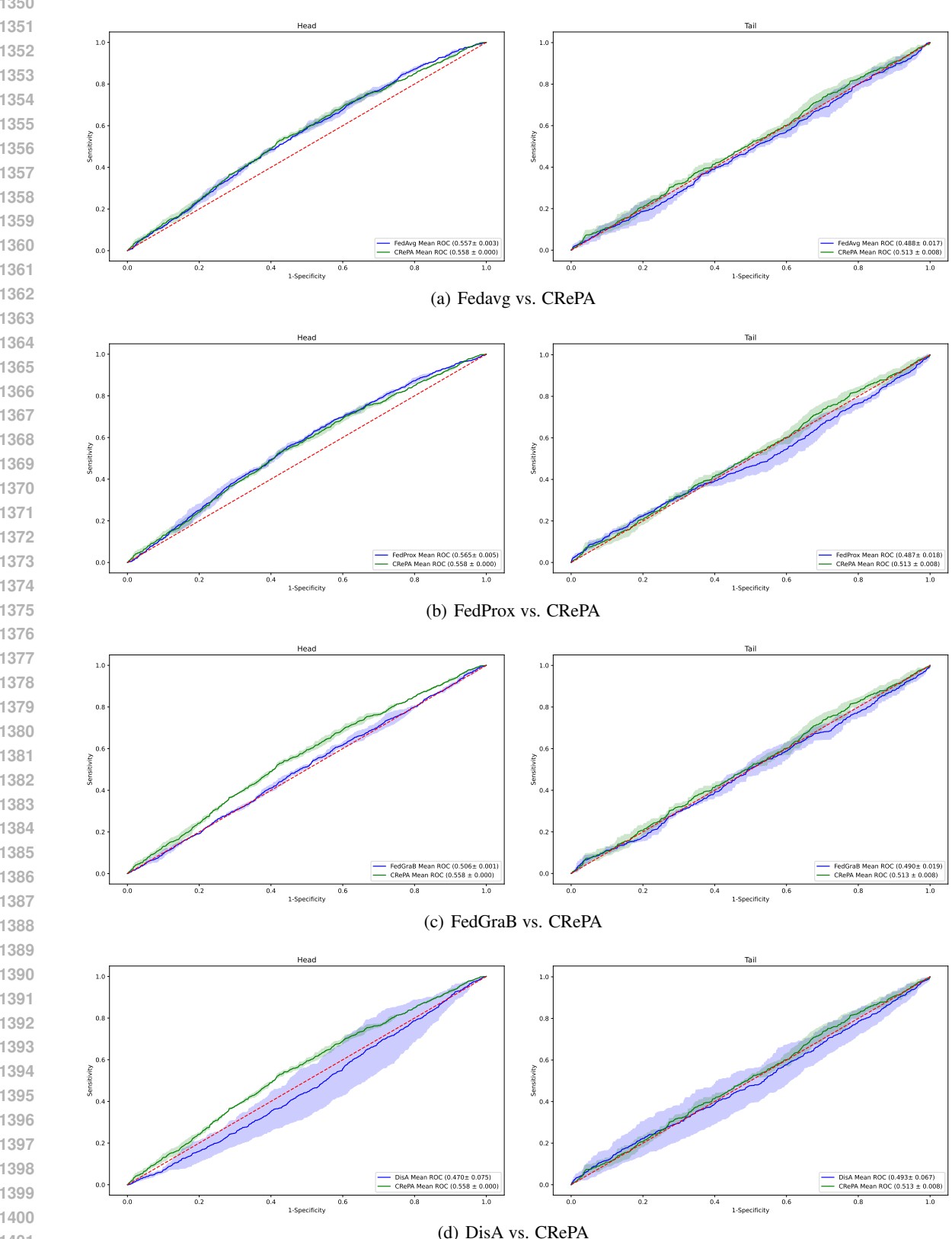

Figure 14: Comparison of ROC Curves Between the Proposed CRePA and Baseline Models.

