# OpenReview forum: "A General Aggregation Federated Learning Intervention Algorithm based on $do$-Calculus"
_ICLR.cc/2025/Conference — Submitted to ICLR 2025_

### Official Review · Reviewer_bqND · 2024-10-28

**Soundness:** 2
**Presentation:** 3
**Contribution:** 3
**Rating:** 5
**Confidence:** 3

**Summary:**

This paper proposes two methods, CRePA and FedLT-CI, to address the challenges posed by data heterogeneity and long-tail distributions in Federated Learning. CRePA improves the performance of the global model on tail categories by re-weighting client contributions, while FedLT-CI leverages causal inference to analyze each client's causal effect and optimizes model aggregation through intervention strategies. Experimental results show that CRePA and FedLT-CI enhance tail category performance while effectively reducing communication costs.

**Strengths:**

1.The overall quality of the writing is commendable.

2.The illustrations in the paper are highly clear and informative.

3.The proposed method is relatively innovative and is supporte1. In the introduction, the authors mention that some researchers' methods addressing Federated Learning heterogeneity and long-tail issues do not consider the impact of different clients on the aggregated model's performance on tail data from a causal perspective, which is "important". However, the paper does not clarify the importance of this perspective, and it appears to result in only a marginal improvement in the data.

**Weaknesses:**

1. In the introduction, the authors mention that some researchers' methods addressing Federated Learning heterogeneity and long-tail issues do not consider the impact of different clients on the aggregated model's performance on tail data from a causal perspective, which is "important". However, the paper does not clarify the importance of this perspective, and it appears to result in only a marginal improvement in the data.

2.The structure of the Introduction section in this paper is not well-organized. The authors begin by introducing the long-tail problem and data heterogeneity issues, followed immediately by an overview of the proposed CRePA and FedLT-CI methods. However, they subsequently discuss limitations in prior work, such as the failure of existing algorithms to address potential long-tail issues in FL, as well as concerns regarding communication costs and the causal perspective. Presenting these issues in previous work after introducing the proposed methods disrupts the logical flow. Clearly, it would be more coherent to first highlight the limitations of prior studies and then present the authors' methods.

Additionally, the authors claim to have "summarized" the contributions of the paper; however, the description of CRePA and FedLT-CI in the contributions section is even more detailed than in previous sections, introducing elements not mentioned earlier. For instance, the authors refer to an adaptive loss function proposed within CRePA, which was not mentioned previously. This makes the contributions section appear overly verbose, while the prior introduction is too brief.

3.The experimental setup section in this paper does not specify the metrics used, and accuracy is the only evaluation metric applied in the experiments. Consequently, the experimental results appear less convincing. Incorporating the AUC-ROC metric may provide further evidence of the model's performance.

**Questions:**

Q1: Is the causal effect important for addressing data heterogeneity and long-tail distribution?

Q2: Is it necessary to conduct further experiments to elaborate on the issue of communication costs?

---

> ### Author Response · Authors · 2024-11-27
> **Response**
>
> Thanks for your valuable suggestions!
>
> Weaknesses
> 1. We will further clarify this in the revision. The causal perspective helps accurately quantify the contribution of different clients to tail data, avoiding the blind allocation of client weights seen in traditional methods, thus improving performance on tail categories. We have added results from medical datasets in the experimental section, demonstrating performance improvements on tail data (such as rare diseases) and showcasing the key role of causal methods in real-world applications. For details, please refer to Appendix E in the revised manuscript.
>
> 2. We have adjusted the structure of the introduction to ensure better logic and coherence.
> We first discuss data heterogeneity and the long-tail problem in federated learning, followed by an analysis of the limitations of existing research, such as inadequate solutions to the long-tail problem, the lack of a causal perspective, and communication cost issues. Then, we introduce the research motivation and briefly present the innovations of the CRePA and FedLT-CI methods, ensuring that the content is more organized and cohesive.
> Furthermore, we recognize that providing overly detailed descriptions of the method in the contributions section may lead to excessive length and inconsistencies with the earlier parts of the introduction. During the revision, we have modified and optimized the contributions section to ensure that it is concise, highlights the key points, and aligns with the content presented earlier. Please refer to the revised manuscript's Introduction.
>
> 3. In response to your feedback, we have incorporated the AUC-ROC metric into our experiments on the PTB-XL dataset to provide a more comprehensive evaluation of the model's performance. Additionally, we have updated the baselines by including the latest models from 2024. For more details, please refer to Appendix E in the revised manuscript.
>
> Questions:
>
> 1. We believe that causal effects play a crucial role in addressing data heterogeneity and long-tail distribution problems, which is currently a gap in the existing research, as reflected in the following aspects:
> (1) Causal effect analysis quantifies the contribution of different clients to the global model, particularly for tail classes. This helps dynamically adjust the aggregation strategy by excluding clients with minimal impact on the tail performance, thereby enhancing the model's adaptability to tail data.
> (2) The causal perspective provides a new analytical framework that explains the reasons for model performance improvements and optimizes existing federated learning frameworks by embedding intervention mechanisms.
>
> 2. Thank you for the reviewer’s suggestion! We have conducted a theoretical analysis of the communication cost of the proposed method in the revised version. Since our FedLT-CI method selectively prevents certain clients from uploading their updated models or gradients during each communication round, compared to traditional aggregation algorithms that require all selected clients to upload their models, this approach effectively reduces the communication burden. Please refer to the newly added Figure 1 in the Introduction section for further details.
>
> We hope our response will satisfy your questions about adjusting the ranking.

---

### Official Review · Reviewer_FXdP · 2024-11-02

**Soundness:** 3
**Presentation:** 2
**Contribution:** 3
**Rating:** 5
**Confidence:** 4

**Summary:**

This manuscript tackles the challenges of federated long-tail learning (Fed-LT), where clients have heterogeneous data that collectively exhibit a global long-tail distribution. The authors introduce two novel approaches: the Client Re-weighted Prior Analyzer (CRePA) and the Federated Long-Tail Causal Intervention Model (FedLTCI). CRePA improves tail performance while maintaining non-tail performance by learning client-specific weight distributions from gradient information. FedLTCI evaluates causal effects on the global model's tail performance, enhancing interpretability. Experiments on CIFAR-10-LT and CIFAR-100-LT show that CRePA achieves state-of-the-art results.

**Strengths:**

1. The issue of long-tail distribution in federated learning (Fed-LT) is significant and intriguing, highlighting the challenges faced in real-world applications.
2. The incorporation of causal inference in the FedLT-CI model is a novel approach in federated long-tailed learning, whichi enhances interpretability.
3. The methodologies employed demonstrate performance improvements in CIFAR-10/100.

**Weaknesses:**

**Motivation:**

1. The motivation behind utilizing aggregation to assist federated learning for long-tail distributions (Fed-LT) is not clearly articulated. In the Fed-LT context, each client typically possesses very few samples of tail classes, making it difficult to ensure that the aggregated model will achieve a more balanced performance across all classes.
2. The necessity of adopting a causal perspective in line 70 raises questions. In the context of Fed-LT, what are the implications of lacking interpretability? Understanding the consequences of this deficiency could strengthen the argument for the proposed approach.
3. The existing methods for Fed-LT present certain limitations that the proposed FedLT-CI aims to overcome. A clearer comparison of these limitations would enhance the reader's understanding of the contributions of this work.
4. The relationship between CREPA and FEDLT-CI is somewhat ambiguous. A more thorough explanation of how these two methodologies interact or complement each other would provide valuable insight into their combined effectiveness.

**Methods and Experiments:**

1. The citations related to data heterogeneity in federated learning are notably outdated. It is important to reference more recent studies in the Fed-LT to provide a comprehensive background and context.
2. The baseline models employed in the experiments seem to be somewhat dated, and the datasets utilized may lack diversity. A broader selection of baselines and more contemporary datasets would strengthen the experimental validation of the proposed methods.
3. The computational overhead associated with CREPA requires further investigation. Additionally, a discussion of the computational costs involved in implementing FEDLT-CI would be beneficial.
4. There is a concern regarding how to prevent clients with a significant representation of head information (e.g., clients c1, c19, c25, c31) but limited tail samples from being consistently excluded during aggregation. This could lead to a decline in the model's representational capacity for tail classes.

**Writing:**

1. In line 60, the transition marked by "therefore" lacks a clear rationale. A more explicit connection to the preceding content would enhance clarity.
2. The privacy issues referenced in line 67 are not clearly defined. Elaborating on the specific privacy concerns would provide a more comprehensive understanding.
3. The methodology for dataset partitioning needs clarification. Specifically, how do the long-tail and heterogeneous datasets generate?
4. In line 152, the statement that data is categorized into tail and non-tail classes seems inconsistent with the experimental design, which mentions "many middle few." This discrepancy should be addressed to avoid confusion.
5. The title "A General Aggregation Federated Learning Intervention Algorithm Based on do-Calculus" does not clearly indicate its relevance to the federated long-tail learning scenario, which may lead to confusion regarding the paper's specific focus.

**Questions:**

1. What prompted the exploration of aggregation as a means to assist federated long-tail learning (Fed-LT)?
2. Why is it necessary to adopt a causal perspective? In the context of FedLT, what are the consequences of lacking interpretability in Fed-LT scenarios?
3. What limitations do previous FedLT approaches have that necessitate the introduction of FedLT-CI?
4. What is the relationship between CREPA and FEDLT-CI?
5. How can we prevent clients with sufficient head information (e.g., c1, c19, c25, c31) but very few tail samples from being consistently excluded during aggregation? What impact could this have on the model's representational capability?

---

> ### Author Response · Authors · 2024-11-27
>
> Thank you for your suggestions!
>
> Motivation:
>
> 1. Our motivation is to enhance the performance of current FL methods on long-tail (LT) data by introducing interventions and calculating clients' causal effects using gradient information. This has been further emphasized in the Introduction of the revised version.
>
> 2. Our model evaluates the impact of clients on the tail performance of the aggregation model by calculating causal effects. This process can be likened to treating the tail performance of the aggregation model as a disease, where client participation in aggregation is considered as treatment, and the treatment effects are assessed. By removing clients with small or negative causal effects, we improve performance and enhance explainability. This has been further elaborated in the Introduction.
>
> 3. We will highlight the limitations of existing methods and the improvements made by FedLT-CI in the text. The revised paper includes a comparison of steps when Fed-CI is integrated into general FL, with further clarification of our innovations. Please refer to Fig. 1 for details.
>
> 4. Fed-CI enhances tail performance by removing clients with small or negative causal effects, based on causal analysis. Afterward, a weighted aggregation algorithm, such as CRePA, can be applied. We have added a diagram in Fig. 1 to clearly illustrate the relationship between the methods and their application.
>
> Methods and Experiments
>
> 1.We highly value the timeliness of citations and have re-examined and updated the relevant literature. In the revision, we have included recent research on data heterogeneity and its solutions in federated learning, providing a more comprehensive discussion in the context of Fed-LT. Thank you for your helpful suggestions!
>
> 2.We updated the experimental baselines with 2024 models and used the PTB-XL ECG dataset with simulated data from 40 hospitals. Our results show significant improvements in recognizing tail classes (rare diseases) across algorithms, with AUC-ROC evaluations included. Detailed results are in Appendix E, strengthening the experimental validation.
>
> 3.Our model does not change the clients' computational cost but increases the servers' computational cost. We will show more details if it is necessary.
>
> 4.Due to word limit, please refer to the response to question 5
>
> Writing: We have polished our manuscript following your advices.
>
> 1.We will further clarify the logical connection before and after "therefore" in the revised manuscript to ensure the coherence and readability of the content.
>
> 2.Some methods rely on statistical information from the data, such as sample counts or feature distributions. However, this approach is not suitable for FL as it may expose sensitive client information. We have provided further clarification in the revised paper.
>
> 3.In Section 5.1, we provide a detailed explanation of the dataset splitting method, including the use of IF (the ratio of training samples in the largest class to the smallest class) to represent the long-tail distribution degree and the parameter $\alpha$ to quantify data heterogeneity (i.e., the Non-IID degree).
>
> 4. In experimental design, researchers often subdivide the head data into head and middle categories, which may lead to inconsistencies in the description. We have added an explanation of the "many, middle, few" classification in the experimental section.
>
> Questions
>
> 1. The core of the FedLT-CI model lies in using do-Calculus to intervene in the FL aggregation process. It evaluates the impact of client participation or non-participation on the global model's tail performance through causal inference and intervenes in the aggregation of clients. This process is illustrated with a figure in the Introduction of the revised manuscript.
>
> 2. We believe the causal perspective offers a new modeling framework that explains performance improvements and optimizes aggregation by embedding an intervention mechanism. This enhances adaptability and tail performance in long-tail scenarios. By quantifying client contributions to the tail performance and dynamically adjusting participating clients, we aim to propose a general method that improves the algorithm's performance on tail classes.
>
> 3. Ignoring the impact of clients on tail data and the lack of flexibility in aggregation strategies makes the approach difficult to generalize and lacking in interpretability.
>
> 4. CRePA is a weighted aggregation method similar to FedAvg. FedLT-CI enhances the performance of CRePA or other baselines on tail data by excluding clients that negatively impact tail performance. Please refer to Fig. 1 added in the Introduction.
>
> 5. In FL, head data dominates and converges quickly, while tail data gradients are often overshadowed. CRePA increases the weight of tail gradients, while Fed-CI improves tail performance by removing clients with minimal causal effect using causal inference.
>
> We hope our response will satisfy your questions about adjusting the ranking.

---

### Official Review · Reviewer_jmkg · 2024-11-06

**Soundness:** 3
**Presentation:** 2
**Contribution:** 2
**Rating:** 3
**Confidence:** 4

**Summary:**

The paper presents two algorithms to address the challenges of data heterogeneity and long-tail distribution in Federated Learning (FL). The first method, Client Re-weighted Prior Analyzer (CRePA), balances the global model's performance on tail and non-tail categories by learning the prior distribution of weights for each client through gradient information. The second method, Federated Long-Tail Causal Intervention Model (FedLT-CI), computes the causal effects of clients on the global model's performance in the tail and enhances interpretability in FL. Extensive experiments on CIFAR10-LT and CIFAR-100-LT datasets demonstrate that CRePA outperforms other baselines.

**Strengths:**

1. Provides a robust theoretical foundation with detailed derivations for the causal intervention framework.
2. Conducts extensive experiments on CIFAR-10/100-LT across varying imbalance and heterogeneity settings.

**Weaknesses:**

1. The motivation of the paper lacks focus, as it seems to tackle data heterogeneity and the long-tail problem separately.
2. It's uncommon for this reviewer to assess a submission introducing two methods addressing distinct challenges. It gives the impression of combining two papers into one.
3. The connection between the two methods is unclear. CRePA appears unrelated to the do-calculus framework mentioned in the title.
4. The authors need to further explore CAUSAL EFFECT in the context of federated learning and compare with relevant baselines.
5. The experiments are limited to CIFAR datasets. Testing on more diverse datasets, especially real-world federated learning long-tail scenarios, would further validate the findings.

**Questions:**

see weaknesses.

---

> ### Author Response · Authors · 2024-11-27
> **Response**
>
> We appreciated your advice and modified our manuscript.
>
> Weaknesses & Questions:
> 1. Our motivation is to improve the performance of current FL algorithms on long-tail data by introducing interventions in traditional FL and calculating the causal effects of clients using gradient information. We have further strengthened the motivation statement in the Introduction. In FL, data heterogeneity and long-tail distribution are interrelated. Heterogeneity manifests as differences in data distribution across clients, such as how patient characteristics may vary between hospitals due to geographic and medical factors. Long-tail distribution refers to the scarcity of samples for certain categories in the global dataset, such as rare diseases in healthcare scenarios. We will further strengthen the description of these two issues in the text. We hope our response meets your satisfaction.
> 2. CRePA treats client weights as a distribution, similar to traditional federated learning methods. Fed-CI is the core innovation of this paper, optimizing the aggregation process through causal intervention by dynamically removing certain clients to improve tail data performance. Fed-CI can be seen as a plugin for federated learning, significantly enhancing the performance of any algorithm on tail data (details can be found in Figure 1 of the revised version). The introduction of these two algorithms not only demonstrates the versatility of Fed-CI but also shows that existing weighting methods can infer the weight distribution and achieve better results.
> 3. Fed-CI can seamlessly integrate with other federated learning algorithms (including CRePA). Its core objective is to improve tail performance by using causal analysis to remove clients that have minimal or negative effects on the tail. After removing such clients, a weighted aggregation method can be directly applied to the remaining clients. In the revised version of the paper, we have added a diagram (Figure 1) to clearly illustrate the logical relationship and application scenarios of the two methods, further enhancing the reader's understanding of this section.
> 4. We design a causal model using client gradients to evaluate their causal effects in the aggregation process, where a larger effect indicates a more significant contribution. Causal analysis is applied to intervene in the aggregation process to improve the algorithm's performance on tail data, which is a novel approach in the current literature. We have added experiments on real medical datasets, included a new 2024 baseline, and incorporated ROC curve evaluation (see Appendix E) to validate the effectiveness of the method.
> 5. We simulated the data distribution from 40 hospitals on the publicly available electrocardiogram dataset PTB-XL and conducted experimental validation in the revised version. By evaluating the model with various metrics, the results show that our method significantly improves the recognition accuracy of tail categories (such as rare diseases) compared to all baselines. This further validates the effectiveness and broad applicability of the method in real-world healthcare scenarios, particularly in the presence of data heterogeneity and long-tail distributions. Detailed information about the experiment can be found in Appendix E of the revised manuscript.
>
> We hope our response will satisfy your questions about adjusting the ranking.

---

### Author Response · Authors · 2024-11-27
**New version**

Dear AC and Reviewers,

Following the reviewers' comments, we have responded to the questions and revised our manuscript. We appreciate your valuable work.

Based on the suggestions, we added the latest 2024 baseline model as a comparison to ensure the research remains cutting-edge and comprehensive. Additionally, we introduced the new PTB-XL medical dataset and conducted a thorough evaluation of the model on this dataset. Furthermore, to more comprehensively measure the model's performance, we incorporated the AUC-ROC metric in our experiments. For detailed information, please refer to Appendix E.

Best,

Yang

---

### Meta-Review · Area_Chair_aqbA · 2024-12-19

**Metareview:**

The submission received the ratings of three reviewers, which recommended 3, 5 and 5, averaging 4.33. Given the plenty of competitive submissions in ICLR, this stands at a score below the borderline. Overall, it seems that the motivation, settings and content organization of this submission all induce some potential points of unclarity, making the reviewers not satisfied with the current version. After rebuttal, the reviewers still do not change their ratings or champion this submission after the AC's reminding. Based on the reviewers' comments and the authors' rebuttal, I thus tend to recommend rejection towards the current submission, and hope the challenging review helps the further improvement of the submission.

Your AC

**Additional Comments On Reviewer Discussion:**

1. Unclear or to some extent ad-hoc motivation.

Not well addressed.

2. Some technical parts require detailed explanation or study.

Not well addressed.

3. Insufficient experiments.

---

### Decision · Program_Chairs · 2025-01-22

Reject